# Beyond the Signs: Nonparametric Tensor Completion via Sign Series

**Chanwoo Lee**
Department of Statistics
University of Wisconsin-Madison
`chanwoo.lee@wisc.edu`

**Miaoyan Wang**
Department of Statistics
University of Wisconsin-Madison
`miaoyan.wang@wisc.edu`

## Abstract

We consider the problem of tensor estimation from noisy observations with possibly missing entries. A nonparametric approach to tensor completion is developed based on a new model which we coin as sign representable tensors. The model represents the signal tensor of interest using a series of structured sign tensors. Unlike earlier methods, the sign series representation effectively addresses both low- and high-rank signals, while encompassing many existing tensor models— including CP models, Tucker models, single index models, structured tensors with repeating entries—as special cases. We provably reduce the tensor estimation problem to a series of structured classification tasks, and we develop a learning reduction machinery to empower existing low-rank tensor algorithms for more challenging high-rank estimation. Excess risk bounds, estimation errors, and sample complexities are established. We demonstrate the outperformance of our approach over previous methods on two datasets, one on human brain connectivity networks and the other on topic data mining.

## 1  Introduction

Higher-order tensors have recently received much attention in enormous fields including social networks [3], neuroscience [38], and genomics [26]. Tensor methods provide effective representation of the hidden structure in multiway data. In this paper we consider the signal plus noise model,

$$\mathcal{Y} = \Theta + \mathcal{E}, \tag{1}$$

where $\mathcal{Y} \in \mathbb{R}^{d_1 \times \cdots \times d_K}$ is an order-$K$ data tensor, $\Theta$ is an unknown signal tensor of interest, and $\mathcal{E}$ is a noise tensor. Our goal is to accurately estimate $\Theta$ from the incomplete, noisy observation of $\mathcal{Y}$. In particular, we focus on the following two problems:

Q1 [Nonparametric tensor estimation]. How to flexibly estimate $\Theta$ under a wide range of structures, including both low-rankness and high-rankness?

Q2 [Complexity of tensor completion]. How many observed tensor entries do we need to consistently estimate the signal $\Theta$?

**Inadequacies of low-rank models.** The signal plus noise model (2) is popular in tensor literature. Existing methods estimate the signal tensor based on low-rankness of $\Theta$ [28, 34]. Common low-rank models include Canonical Polyadic (CP) tensors [24], Tucker tensors [11], and block tensors [40]. While these methods have shown great success in theory, tensors in applications often violate the low-rankness. Here we provide two examples to illustrate the limitation of classical models.

The first example reveals the sensitivity of tensor rank to order-preserving transformations. Let $\mathcal{Z} \in \mathbb{R}^{30 \times 30 \times 30}$ be an order-3 tensor with tensor rank$(\mathcal{Z}) = 3$ (formal definition is deferred to the end of this section). Suppose a monotonic transformation $f(z) = (1 + \exp(-cz))^{-1}$ is applied to $\mathcal{Z}$

entrywise, and we let the signal $\Theta$ in model (1) be the tensor after transformation. Figure 1a plots the numerical rank (see Appendix B.1) of $\Theta$ versus $c$. As we see, the rank increases rapidly with $c$, rendering traditional low-rank tensor methods ineffective in the presence of mild order-preserving nonlinearities. Similar observations apply to both CP and Tucker models, and more generally, to low-rank models with scale-sensitive rank measures. In digital processing [17] and genomics analysis [26], the tensor of interest often undergoes unknown transformation prior to measurements. The sensitivity to transformation makes the low-rank model less desirable in practice.

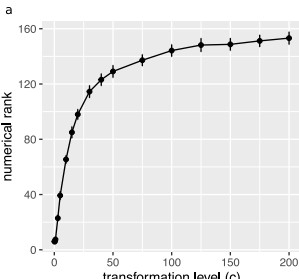
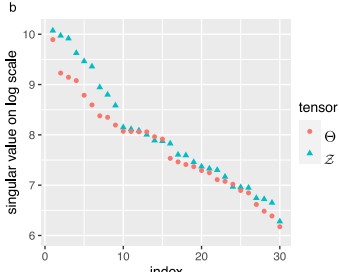

Figure 1: (a) Numerical tensor rank of $\Theta$ vs. transformation level $c$ in the first example. (b) Top $d = 30$ tensor singular values in the second example.

The second example demonstrates the inadequacy of classical low-rankness in representing special structures. Here we consider the signal tensor of the form $\Theta = \log(1 + \mathcal{Z})$, where $\mathcal{Z} \in \mathbb{R}^{d \times d \times d}$ is an order-3 tensor with entries $\mathcal{Z}(i, j, k) = d^{-1} \max(i, j, k)$ for $i, j, k \in \{1, \ldots, d\}$. The matrix analogy of $\Theta$ was studied in graphon analysis [8]. In this case neither $\Theta$ nor $\mathcal{Z}$ is low-rank; in fact, the rank is no smaller than the dimension $d$ as illustrated in Figure 1b. Again, classical low-rank models fail to address this type of tensor structure.

In the above and many other examples, the signal tensors $\Theta$ of interest have high rank. Classical low-rank models will miss these important structures. The observations have motivated us to develop more flexible tensor modeling.

**Our contributions.** We develop a new model called sign representable tensors to address the aforementioned challenges. Figure 2 illustrates our main idea. Our approach is built on the sign series representation of the signal tensor, and we propose to estimate the sign tensors through a series of weighted classifications. In contrast to existing methods, our method is guaranteed to recover a wide range of low- and high-rank signals. We highlight two main contributions that set our work apart from earlier literature.

Statistically, the problem of high-rank tensor estimation is challenging. Existing estimation theory [3, 34, 7] exclusively focuses on the regime of fixed $r$ and growing $d$. However, such premise fails in high-rank tensors, where the rank may grow with, or even exceed, the dimension. A proper notion of nonparametric complexity is crucial. We show that, somewhat surprisingly, the sign tensor series not only preserves all information in the original signals, but also brings the benefits of flexibility and accuracy over classical low-rank models. The results fill the gap between parametric (low-rank) and nonparametric (high-rank) tensors, thereby greatly enriching the tensor model literature.

Computationally, a number of polynomial-time algorithms are readily available under moderate-to-high signal-to-noise ratio for 1-bit tensor estimation [39, 20, 17]. These algorithms enjoy computational efficiency while being restricted to binary inputs. Our work is orthogonal to these algorithm development, and we show that the high-rank tensor estimate is provably reducible to a series of binary tensor problems with carefully-designed weights. This reduction provides a generic engine to empower existing algorithms for a wider range of structured tensor problems. We use a divide-and-concur approach to combine efficient base algorithms, thereby achieving computational accuracy without the need to reinvent the wheel. The flexibility to import and adapt existing tensor algorithms is one advantage of our method.

We also highlight the challenges associated with tensors compared to matrices. High-rank matrix estimation has been studied under graphon models [41, 44, 8], nonlinear models [15], and subspace clustering [35, 13]. In particular, the recent work [30] proposes a general nonparametric framework to address a variety of matrix problems including regression and completion. However, high-rank tensor problems is more challenging, because the tensor rank often exceeds the dimension when

order $K$ greater than two [4]. This is in sharp contrast to matrices ($K = 2$). We show that, applying matrix methods to higher-order tensors results in suboptimal estimates. A full exploitation of the higher-order structure is needed; this is another challenge we address in this paper.

**Notation.** We use $[n] = \{1, \ldots, n\}$ for $n$-set with $n \in \mathbb{N}_+$, $a_n \lesssim b_n$ if $\lim_{n \to \infty} a_n/b_n \leq c$ for some constant $c > 0$, and $a_n \asymp b_n$ if $c_1 \leq \lim_{n \to \infty} a_n/b_n \leq c_2$ for some constants $c_1, c_2 > 0$. We use $\mathcal{O}(\cdot)$ to denote the big-O notation, $\tilde{\mathcal{O}}(\cdot)$ the variant hiding logarithmic factors. Let $\Theta \in \mathbb{R}^{d_1 \times \cdots \times d_K}$ denote an order-$K$ $(d_1, \ldots, d_K)$-dimensional tensor, and $\Theta(\omega) \in \mathbb{R}$ denote the tensor entry indexed by $\omega \in [d_1] \times \cdots \times [d_K]$. An event $A$ is said to occur "with very high probability" if $\mathbb{P}(A)$ tends to 1 faster than any polynomial of tensor dimension $d := \min_k d_k \to \infty$. The tensor rank [24] is defined by $\text{rank}(\Theta) = \min\{r \in \mathbb{N} : \Theta = \sum_{s=1}^{r} \boldsymbol{a}_s^{(1)} \otimes \cdots \otimes \boldsymbol{a}_s^{(K)}\}$, where $\boldsymbol{a}_s^{(k)} \in \mathbb{R}^{d_k}$ are vectors for $k \in [K], s \in [r]$, and $\otimes$ denotes the outer product of vectors. We use $\text{sgn}(\cdot) : \mathbb{R} \to \{-1, 1\}$ to denote the sign function, where $\text{sgn}(y) = 1$ if $y \geq 0$ and $-1$ otherwise. We allow univariate functions, such as $\text{sgn}(\cdot)$ and general $f : \mathbb{R} \to \mathbb{R}$, to be applied to tensors in an element-wise manner.

## 2 Model and proposal overview

Let $\mathcal{Y}$ be an order-$K$ $(d_1, \ldots, d_K)$-dimensional tensor generated from the model

$$\mathcal{Y} = \Theta + \mathcal{E}, \tag{2}$$

where $\Theta \in \mathbb{R}^{d_1 \times \cdots \times d_K}$ is an unknown signal tensor, and $\mathcal{E}$ is a noise tensor consisting of zero-mean, independent but not necessarily identically distributed entries. We allow heterogenous noise, in that the marginal distribution of noise entry $\mathcal{E}(\omega)$ may depend on $\omega$. For a cleaner exposition, we assume the noise is bounded and the range of observation is in $[-1, 1]$; the extension to unbounded observations with sub-Gaussian noise is provided in Appendix B.3. Our observation is an incomplete data tensor from (2), denoted $\mathcal{Y}_\Omega$, where $\Omega \subset [d_1] \times \cdots \times [d_K]$ is the index set of observed entries. We consider a general model on $\Omega$ that allows both uniform and non-uniform samplings. Specifically, let $\Pi = \{p_\omega\}$ be an arbitrarily predefined probability distribution over the full index set with $\sum_{\omega \in [d_1] \times \cdots \times [d_K]} p_\omega = 1$. We use $\omega \sim \Pi$ to denote the sampling rule, meaning $\omega$ in $\Omega$ are i.i.d. draws with replacement from distribution $\Pi$. The goal is to estimate $\Theta$ from $Y_\Omega$. Note that $\Theta$ is not necessarily low-rank.

**Proposal intuition.** Before describing our main results, we provide the intuition behind our method. In the two examples in Section 1, the high-rankness in the signal $\Theta$ makes the estimation challenging. Now let us examine the sign of the $\pi$-shifted signal $\text{sgn}(\Theta - \pi)$ for any given $\pi \in [-1, 1]$. It turns out that, these sign tensors share the same sign patterns as low-rank tensors. Indeed, the signal tensor in the first example has the same sign pattern as a rank-4 tensor, since $\text{sgn}(\Theta - \pi) = \text{sgn}(\mathcal{Z} - f^{-1}(\pi))$. The signal tensor in the second example has the same sign pattern as a rank-2 tensor, since $\text{sgn}(\Theta(i, j, k) - \pi) = \text{sgn}(\max(i, j, k) - d(e^\pi - 1))$ (see Example 5 in Section 3).

The above observation suggests a general framework to estimate both low- and high-rank signal tensors. Figure 2 illustrates the main crux of our method. We propose to estimate the signal tensor $\Theta$ by taking the average over structured sign tensors

$$\hat{\Theta} = \frac{1}{2H + 1} \sum_{\pi \in \mathcal{H}} \text{sgn}(\hat{\mathcal{Z}}_\pi), \text{ where } \hat{\mathcal{Z}}_\pi = \underset{\text{low rank tensor } \mathcal{Z}}{\arg \min} \text{Weighted-Loss}(\text{sgn}(\mathcal{Z}), \text{sgn}(\mathcal{Y}_\Omega - \pi)). \tag{3}$$

Here $\text{sgn}(\hat{\mathcal{Z}}_\pi) \in \{-1, 1\}^{d_1 \times \cdots \times d_K}$ is the sign tensor estimated at a series of $\pi \in \mathcal{H} = \{-1, \ldots, -1/H, 0, 1/H, \ldots, 1\}$, and Weighted-Loss$(\cdot, \cdot)$ denotes a classification objective function with an entry-specific weight to each tensor entry; its specific form will be described in Section 3.2. To obtain $\text{sgn}(\hat{\mathcal{Z}}_\pi)$ for a given $\pi$, we propose to dichotomize the data tensor into a sign tensor $\text{sgn}(\mathcal{Y}_\Omega - \pi)$ and estimate the de-noised sign by performing weighted classification.

Our approach is built on the nonparametric sign representation of signal tensors. We show that a careful aggregation of dichotomized data preserves all information in the original signals and brings benefits of accuracy and flexibility over classical low-rank models. Unlike traditional methods, the sign representation is guaranteed to recover both low- and high-rank signals. In addition, a total of $H = \text{poly}(d)$ dichotomized problems suffice to recover $\Theta$ under the considered model. The method therefore enjoys both statistical effectiveness and computational efficiency.

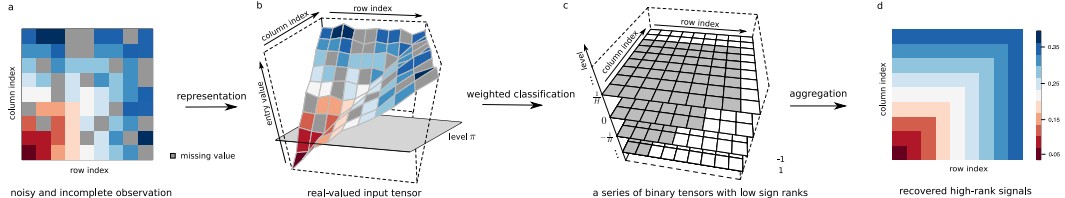

Figure 2: Illustration of our method in the context of an order-2 tensor (a.k.a. matrix). (a): a noisy, incomplete tensor input. (b)-(c): estimation of sign tensor series $\text{sgn}(\Theta - \pi)$ for $\pi \in \{-1, \ldots, -1/H, 0, 1/H, \ldots, 1\}$. (d): recovered signal $\hat{\Theta}$. The depicted signal is a full-rank matrix based on Example 5 in Section 3.

## 3  Oracle properties of sign representable tensors

This section develops sign representable tensor models for $\Theta$ in (2). We characterize the algebraic and statistical properties of sign tensor series, which serves the foundation for our method.

### 3.1  Sign-rank and sign tensor series

Let $\Theta$ be the tensor of interest, and $\text{sgn}(\Theta)$ the corresponding sign pattern. The sign patterns induce an equivalence relationship between tensors. Two tensors are called sign equivalent, denoted $\simeq$, if they have the same sign pattern.

**Definition 1** (Sign-rank). The sign-rank of a tensor $\Theta \in \mathbb{R}^{d_1 \times \cdots \times d_K}$ is defined by the minimal rank among all tensors that share the same sign pattern as $\Theta$; i.e.,

$$\text{srank}(\Theta) = \min\{\text{rank}(\Theta') \colon \Theta' \simeq \Theta, \ \Theta' \in \mathbb{R}^{d_1 \times \cdots \times d_K}\}.$$

This concept is important in combinatorics [10], complexity theory [1], and quantum mechanics [12]; we extend the notion to continuous-valued tensors. Note that the sign-rank concerns only the sign pattern but discards the magnitude information of $\Theta$. In particular, $\text{srank}(\Theta) = \text{srank}(\text{sgn}\Theta)$.

Like most tensor problems [23], determining the sign-rank is NP hard in the worst case [1]. Fortunately, tensors arisen in applications often possess special structures that facilitate analysis. The sign-rank is upper bounded by tensor rank. More generally, we show the following properties.

**Proposition 1** (Upper bounds of the sign-rank).
(a) [Upper bounds] For any strictly monotonic function $g \colon \mathbb{R} \to \mathbb{R}$ with $g(0) = 0$, we have $\text{srank}(\Theta) \leq \text{rank}(g(\Theta))$.
(b) [Broadness] For every order $K \geq 2$ and dimension $d$, there exist tensors $\Theta \in \mathbb{R}^{d \times \cdots \times d}$ such that $\text{rank}(\Theta) \geq d$ but $\text{srank}(\Theta - \pi) \leq 2$ for all $\pi \in \mathbb{R}$.

Propositions 1 demonstrates the strict broadness of low sign-rank family over the usual low-rank family. In particular, the sign-rank can be much smaller than the tensor rank, as we have shown in the two examples of Section 1. We provide several additional examples in Appendix B.2 in which the tensor rank grows with dimension $d$ but the sign-rank remains a constant. The results highlight the advantages of using sign-rank in the high-dimensional tensor analysis.

We now introduce a tensor family, which we coin as "sign representable tensors".

**Definition 2** (Sign representable tensors). Fix a level $\pi \in [-1, 1]$. A tensor $\Theta$ is called $(r, \pi)$-sign representable, if the tensor $(\Theta - \pi)$ has sign-rank bounded by $r$. A tensor $\Theta$ is called $r$-sign (globally) representable, if $\Theta$ is $(r, \pi)$-sign representable for all $\pi \in [-1, 1]$. The collection $\{\text{sgn}(\Theta - \pi) \colon \pi \in [-1, 1]\}$ is called the sign tensor series. We use $\mathscr{P}_{\text{sgn}}(r) = \{\Theta \colon \max_{\pi \in [-1,1]} \text{srank}(\Theta - \pi) \leq r\}$ to denote the $r$-sign representable tensor family.

We next show that the $r$-sign representable tensor family is a general model that incorporates most existing tensor models, including low-rank tensors, single index models, GLM models, and structured tensors with repeating entries.

**Example 1** (CP/Tucker low-rank models). The CP and Tucker low-rank tensors are the two most popular tensor models [29]. Let $\Theta$ be a low-rank tensor with CP rank $r$. We see that $\Theta$ belongs to the sign representable family; i.e., $\Theta \in \mathscr{P}_{\text{sgn}}(r + 1)$ (the constant 1 is due to $\text{rank}(\Theta - \pi) \leq r + 1$).

Similar results hold for Tucker low-rank tensors $\Theta \in \mathscr{P}_{\text{sgn}}(r+1)$, where $r = \prod_k r_k$ with $r_k$ being the $k$-th mode Tucker rank of $\Theta$.

**Example 2** (Tensor block models (TBMs)). Tensor block model [40, 9] assumes a checkerboard structure among tensor entries under marginal index permutation. The signal tensor $\Theta$ takes at most $r$ distinct values, where $r$ is the total number of multiway blocks. Our model incorporates TBM because $\Theta \in \mathscr{P}_{\text{sgn}}(r)$.

**Example 3** (Generalized linear models (GLMs)). Let $\mathcal{Y}$ be a binary tensor from a logistic model [39] with mean $\Theta = \text{logit}(\mathcal{Z})$, where $\mathcal{Z}$ is a latent low-rank tensor. Notice that $\Theta$ itself may be high-rank (see Figure 1a). By definition, $\Theta$ is a low-rank sign representable tensor. Same conclusion holds for general exponential-family models with a (known) link function [25].

**Example 4** (Single index models (SIMs)). Single index model is a flexible semiparametric model proposed in economics [36] and high-dimensional statistics [5, 14]. The SIM assumes the existence of a (unknown) monotonic function $g\colon \mathbb{R} \to \mathbb{R}$ such that $g(\Theta)$ has rank $r$. We see that $\Theta$ belongs to the sign representable family; i.e., $\Theta \in \mathscr{P}_{\text{sgn}}(r+1)$.

**Example 5** (Structured tensors with repeating entries). Here we revisit the model introduced in Figure 1b of Section 1. Let $\Theta$ be an order-$K$ tensor with entries $\Theta(i_1, \ldots, i_K) = \log(1 + \max_k x_{i_k}^{(k)})$, where $x_{i_k}^{(k)}$ are given numbers in $[0, 1]$ for all $i_k \in [d_k], k \in [K]$. We conclude that $\Theta \in \mathscr{P}_{\text{sgn}}(2)$, because the sign tensor $\text{sgn}(\Theta - \pi)$ with an arbitrary $\pi \in (0, \log 2)$ is a block tensor with at most two blocks (see Figure 2c). Similar results extend to structured tensors with entries $\Theta(i_1, \ldots, i_K) = g(\max_k x_{i_k}^{(k)})$, where $g(\cdot)$ is a polynomial of degree $r$. In this case, $\Theta$ is a high-rank tensor with at most $d_{\max}$ distinct entries but we have $\Theta \in \mathscr{P}_{\text{sgn}}(2r)$ (see proofs in Appendix B.2). These structured tensors are related to hypergraphons [45, 33]. We discuss the connection in Appendix B.2.

## 3.2 Statistical characterization of sign tensors via weighted classification

We now provide the explicit form of the weighted loss introduced in (3), and show that sign tensors are characterized by weighted classification. The results bridge the algebraic and statistical properties of sign representable tensors.

For a given $\pi \in [-1, 1]$, define a $\pi$-shifted data tensor $\bar{\mathcal{Y}}_\Omega$ with entries $\bar{\mathcal{Y}}(\omega) = (\mathcal{Y}(\omega) - \pi)$ for $\omega \in \Omega$. We propose a weighted classification objective function

$$L(\mathcal{Z}, \bar{\mathcal{Y}}_\Omega) = \frac{1}{|\Omega|} \sum_{\omega \in \Omega} \underbrace{|\bar{\mathcal{Y}}(\omega)|}_{\text{weight}} \times \underbrace{|\text{sgn}\mathcal{Z}(\omega) - \text{sgn}\bar{\mathcal{Y}}(\omega)|}_{\text{classification loss}}, \tag{4}$$

where $\mathcal{Z} \in \mathbb{R}^{d_1 \times \cdots \times d_K}$ is the decision variable to be optimized, $|\bar{\mathcal{Y}}(\omega)|$ is the entry-specific weight equal to the distance from the tensor entry to the target level $\pi$. The entry-specific weights incorporate the magnitude information into classification, where entries far away from the target level are penalized more heavily in the objective. In the special case of binary tensor $\mathcal{Y} \in \{-1, 1\}^{d_1 \times \cdots \times d_K}$ and target level $\pi = 0$, the loss (4) reduces to usual classification loss.

Our proposed weighted classification function (4) is important for characterizing $\text{sgn}(\Theta - \pi)$. Define the weighted classification risk

$$\text{Risk}(\mathcal{Z}) = \mathbb{E}_{\mathcal{Y}_\Omega} L(\mathcal{Z}, \bar{\mathcal{Y}}_\Omega), \tag{5}$$

where the expectation is taken with respect to $\mathcal{Y}_\Omega$ under model (2) and the sampling distribution $\omega \sim \Pi$. The form of $\text{Risk}(\cdot)$ implicitly depends on $\pi$; we suppress $\pi$ when no confusion arises.

**Proposition 2** (Global optimum of weighted risk). Suppose the data $\mathcal{Y}_\Omega$ is generated from model (2) with $\Theta \in \mathscr{P}_{\text{sgn}}(r)$. Then, for all $\bar{\Theta}$ that are sign equivalent to $\text{sgn}(\Theta - \pi)$,

$$\text{Risk}(\bar{\Theta}) = \inf\{\text{Risk}(\mathcal{Z})\colon \mathcal{Z} \in \mathbb{R}^{d_1 \times \cdots \times d_K}\} = \inf\{\text{Risk}(\mathcal{Z})\colon \text{rank}(\mathcal{Z}) \leq r\}.$$

The results show that the sign tensor $\text{sgn}(\Theta - \pi)$ optimizes the weighted classification risk. This fact suggests a practical procedure to estimate $\text{sgn}(\Theta - \pi)$ via empirical risk optimization of $L(\mathcal{Z}, \bar{\mathcal{Y}}_\Omega)$. In order to establish the recovery guarantee, we shall address the uniqueness (up to sign equivalence) for the optimizer of $\text{Risk}(\cdot)$. The local behavior of $\Theta$ around $\pi$ plays a key role in the accuracy.

Some additional notation is needed for stating the results in full generality. Let $d_{\text{total}} = \prod_{k=1}^{K} d_k$ denote the total number of tensor entries, and $\Delta s = 1/d_{\text{total}}$ a small tolerance. We quantify the

distribution of tensor entries $\Theta(\omega)$ using a pseudo density, i.e., histogram with bin width $2\Delta s$. Let $G(\pi) := \mathbb{P}_{\omega \sim \Pi}[\Theta(\omega) \leq \pi]$ denote the cumulative distribution function (CDF) of $\Theta(\omega)$ under $\omega \sim \Pi$. We partition $[-1, 1] = \mathcal{N}^c \cup \mathcal{N}$, such that the pseudo density based on $2\Delta$-bin is uniformly bounded over $\mathcal{N}^c$; i.e,

$$\mathcal{N}^c = \left\{ \pi \in [-1, 1] \colon \frac{G(\pi + \Delta s) - G(\pi - \Delta s)}{\Delta s} \leq C \right\}, \text{ for some universal constant } C > 0,$$

and $\mathcal{N}$ otherwise. Informally, the set $\mathcal{N}$ collects the jump points for which the pseudo density is unbounded. Both $\Theta$ and its induced CDF $G$ implicitly depend on the tensor dimension.

**Assumption 1** ($\alpha$-smoothness). Fix $\pi \in \mathcal{N}^c$. Assume there exist constants $\alpha = \alpha(\pi) > 0, c = c(\pi) > 0$, independent of tensor dimension, such that,

$$\sup_{\Delta s \leq t < \rho(\pi, \mathcal{N})} \frac{G(\pi + t) - G(\pi - t)}{t^\alpha} \leq c, \tag{6}$$

where $\rho(\pi, \mathcal{N}) := \min_{\pi' \in \mathcal{N}} |\pi - \pi'| + \Delta s$ denotes the adjusted distance from $\pi$ to the nearest point in $\mathcal{N}$. We make the convention that $\rho(\pi, \mathcal{N}) = \infty$ when $\mathcal{N} = \emptyset$. The largest possible $\alpha = \alpha(\pi)$ in (6) is called the smoothness index at level $\pi$. We define that $\alpha = \infty$ if the numerator in (6) is zero. A tensor $\Theta$ is called $\alpha$-globally smooth, if (6) holds with global constants $\alpha > 0, c > 0$ for all $\pi \in \mathcal{N}^c$. A similar notion of $\alpha$-smoothness was previously developed in different contexts of nonparametric function estimation; see Figures 3 in Lee et al [30] for an illustration.

The smoothness index $\alpha$ quantifies the intrinsic hardness of recovering $\text{sgn}(\Theta - \pi)$ from $\text{Risk}(\cdot)$. The value of $\alpha$ depends on both the sampling distribution $\omega \sim \Pi$ and the behavior of $\Theta(\omega)$. The recovery is easier at levels where points are less concentrated around $\pi$ with a large value of $\alpha > 1$, or equivalently, when $G(\pi)$ remains almost flat around $\pi$. A small value of $\alpha < 1$ indicates the nonexistent (infinite) density at level $\pi$, or equivalently, when the $G(\pi)$ jumps by greater than the tolerance $\Delta s$ at $\pi$. Table 2 illustrates the $G(\pi)$ for various models of $\Theta$ (see Section 5).

We now reach the main theorem in this section. For two tensors $\Theta_1, \Theta_2$, define the mean absolute error (MAE) as $\text{MAE}(\Theta_1, \Theta_2) = \mathbb{E}_{\omega \sim \Pi} |\Theta_1(\omega) - \Theta_2(\omega)|$.

**Theorem 1** (Identifiability). Assume $\Theta \in \mathscr{P}_{\text{sgn}}(r)$ is $\alpha$-globally smooth. Then, for all $\pi \in \mathcal{N}^c$ and tensors $\bar{\Theta} \simeq \text{sgn}(\Theta - \pi)$, we have

$$\text{MAE}(\text{sgn}\mathcal{Z}, \text{sgn}\bar{\Theta}) \lesssim C(\pi) \left[ \text{Risk}(\mathcal{Z}) - \text{Risk}(\bar{\Theta}) \right]^{\alpha/(\alpha+1)} + \Delta s, \quad \text{for all } \mathcal{Z} \in \mathbb{R}^{d_1 \times \cdots \times d_K},$$

where $C(\pi) > 0$ is independent of $\mathcal{Z}$.

The result establishes the recovery stability of sign tensors $\text{sgn}(\Theta - \pi)$ using optimization with population risk (5). The bound immediately shows the uniqueness of the optimizer for $\text{Risk}(\cdot)$ up to a $\Delta s$-measure set under $\Pi$. We find that a higher value of $\alpha$ implies more stable recovery, as intuition would suggest. Similar results hold for optimization with sample risk (4) (see Section 4).

We conclude this section by applying Assumption 1 to the examples described in Section 3.1. For simplicity, suppose $\Pi$ is the uniform sampling. The tensor block model is $\infty$-globally smooth. This is because the set $\mathcal{N}$ consists of finite $2\Delta s$-bin's covering the distinct block means in $\Theta$. Furthermore, we have $\alpha = \infty$ for all $\pi \in \mathcal{N}^c$, since the numerator in (6) is zero. Similarly, the high-rank $(d, d, d)$-dimensional tensor $\Theta(i, j, k) = \log(1 + \frac{1}{d} \max(i, j, k))$ is $\infty$-globally smooth because $\alpha = \infty$ for all $\pi$ except those in $\mathcal{N}$, where $\mathcal{N}$ collects $d$ many $2\Delta s$-bin's covering $\log(1 + i/d)$ for all $i \in [d]$.

## 4 Nonparametric tensor completion via sign series

In previous sections we have established the sign series representation and its relationship to classification. In this section, we present our learning reduction proposal in details (Figure 2). We provide the estimation error bound and address the empirical implementation of the method.

### 4.1 Statistical error and sample complexity

Given a noisy incomplete tensor observation $\mathcal{Y}_\Omega$ from model (2), we cast the problem of estimating $\Theta$ into a series of weighted classifications. Specifically, we propose the signal tensor estimate using

averaged structured sign tensors

$$\hat{\Theta} = \frac{1}{2H+1} \sum_{\pi \in \mathcal{H}} \operatorname{sgn} \hat{\mathcal{Z}}_{\pi}, \quad \text{with} \quad \hat{\mathcal{Z}}_{\pi} = \underset{\mathcal{Z}: \operatorname{rank}\mathcal{Z} \leq r}{\arg\min} L(\mathcal{Z}, \mathcal{Y}_{\Omega} - \pi), \tag{7}$$

where $\mathcal{H} = \{-1, \ldots, -1/H, 0, 1/H, \ldots, 1\}$ is the series of levels to aggregate, $L(\cdot, \cdot)$ denotes the weighted classification objective defined in (4), and the rank constraint on $\mathcal{Z}$ follows from Proposition 2. For the theory, we assume the true $r$ is known; in practice, $r$ could be chosen in a data adaptive fashion via cross-validation or elbow method [21].

The next theorem establishes the statistical convergence for the sign tensor estimate (7).

**Theorem 2** (Sign tensor estimation). Suppose $\Theta \in \mathscr{P}_{\operatorname{sgn}}(r)$ and $\Theta(\omega)$ is $\alpha$-globally smooth under $\omega \sim \Pi$. Let $\hat{\mathcal{Z}}_{\pi}$ be the estimate in (7), $d = \max_{k \in [K]} d_k$, and $t_d = \frac{dr \log |\Omega|}{|\Omega|} \lesssim 1$. Then, for all $\pi \in \mathcal{N}^c$, with very high probability over $\mathcal{Y}_{\Omega}$,

$$\operatorname{MAE}(\operatorname{sgn}\hat{\mathcal{Z}}_{\pi}, \operatorname{sgn}(\Theta - \pi)) \lesssim t_d^{\alpha/(\alpha+2)} + \frac{1}{\rho^2(\pi, \mathcal{N})} t_d. \tag{8}$$

Theorem 2 provides the error bound for the sign tensor estimation. Compared to the population results in Theorem 1, we explicitly reveal the dependence of accuracy on the sample complexity and the level $\pi$. The result demonstrates the polynomial decay of sign errors with $|\Omega|$. Our sign estimate achieves consistent recovery using as few as $\tilde{\mathcal{O}}(dr)$ noisy entries.

Recall that $\mathcal{N}$ collects the levels for which the sign tensor is possibly nonrecoverable. Let $|\mathcal{N}|$ be the covering number of $\mathcal{N}$ with $2\Delta s$-bin's, i.e, $|\mathcal{N}| = \lceil \operatorname{Leb}(\mathcal{N})/2\Delta s \rceil$, where $\operatorname{Leb}(\cdot)$ is the Lebesgue measure and $\lceil \cdot \rceil$ is the ceiling function. Combining the sign representability of the signal tensor and the sign estimation accuracy, we obtain our main results on nonparametric tensor estimation.

**Theorem 3** (Tensor estimation error). Consider the same conditions of Theorem 2. Let $\hat{\Theta}$ be the estimate in (7). For any resolution parameter $H \in \mathbb{N}_+$, with very high probability over $\mathcal{Y}_{\Omega}$,

$$\operatorname{MAE}(\hat{\Theta}, \Theta) \lesssim (t_d \log H)^{\frac{\alpha}{\alpha+2}} + \frac{1 + |\mathcal{N}|}{H} + t_d H \log H. \tag{9}$$

In particular, setting $H = (1 + |\mathcal{N}|)^{1/2} t_d^{-1/2} \asymp \operatorname{poly}(d)$ yields the tightest upper bound in (9).

Theorem 3 demonstrates the convergence rate of our tensor estimation. The bound (9) reveals three sources of errors: the estimation error for sign tensors, the bias from sign series representations, and the variance thereof. The resolution parameter $H$ controls the bias-variance tradeoff. We remark that the signal estimation error (9) is generally no better than the corresponding sign error (8). This is to be expected, since magnitude estimation is a harder problem than sign estimation.

In the special case of full observation with equal dimension $d_1 = \cdots = d_K = d$ and bounded $|\mathcal{N}| \leq C$, our signal estimate achieves convergence

$$\operatorname{MAE}(\hat{\Theta}, \Theta) \lesssim rd^{-(K-1)\min(\frac{\alpha}{\alpha+2}, \frac{1}{2})} \log^2 d,$$

by setting $H \asymp d^{(K-1)/2}$. Compared to earlier methods, our estimation accuracy applies to both low- and high-rank signal tensors. The rate depends on the sign complexity $\Theta \in \mathscr{P}_{\operatorname{sgn}}(r)$, and this $r$ is often much smaller than the usual tensor rank (see Section 3.1). Our result also reveals that the convergence becomes favorable as the order of data tensor increases.

We apply our general theorem to the main examples in Section 3.1, and we compare the results with existing literature (Table 1). The numerical comparison is provided in Section 5.

**Example 2** (TBMs). Consider a tensor block model with $r$ multiway blocks. Our result implies a rate $\tilde{\mathcal{O}}(d^{-(K-1)/2})$ by taking $\alpha = \infty$ and $|\mathcal{N}| \leq r^K \lesssim \mathcal{O}(1)$. This rate agrees with the previous root-mean-square error (RMSE) for block tensor estimation [40].

**Example 3** (GLMs). Consider a GLM tensor $\Theta = g(\mathcal{Z})$, where $g$ is a known link function and $\mathcal{Z}$ is a latent low-rank tensor. Suppose the CDF of $\Theta(\omega)$ is uniformly bounded as $d \to \infty$. Applying our results with $\alpha = 1$ and finite $|\mathcal{N}|$ yields $\tilde{\mathcal{O}}(d^{-(K-1)/3})$. This rate is slightly slower than the parametric RMSE rate [43, 39, 27], as expected. The reason is that our estimate remains valid for unknown $g$ and general high-rank tensors. The nonparametric rate is the price one has to pay for not knowing the form $\Theta = g(\mathcal{Z})$ as a priori.

Table 1: Summary of our statistical rates compared to existing works under different models. For notational simplicity, we present error rates assuming equal tensor dimension in all modes and finite $|\mathcal{N}|$ for the smooth tensor model. Here $K$ denotes tensor order and $d$ denotes tensor dimension.

| Model | $\alpha$ | $|\mathcal{N}|$ | Our rate (power of $d$) | Comparison with previous result |
|---|---|---|---|---|
| Tensor block model | $\infty$ | Finite | $-(K-1)/2$ | Achieves minimax rate [40]. |
| Single index model | 1 | 0 | $-(K-1)/3$ | Not available for general $K > 3$; Improves previous rate -1/4 for $K = 2$ [15]. |
| Generalized linear model | 1 | 0 | $-(K-1)/3$ | Close to minimax rate [43, 39, 27]. |
| Structure with repeating entries | $\infty$ | $d$ | $-(K-2)/2$ | Not available. |
| Smooth tensor | $\alpha$ | Finite | $-(K-2)\min(\frac{\alpha}{\alpha+2}, \frac{1}{2})$ | Not available. |

**Example 4** (SIMs). The earlier example has shown the nonparametric rate $\tilde{\mathcal{O}}(d^{-(K-1)/3})$ when applying our method to single index tensor model. In the matrix case with $K = 2$, our theorem yields error rate $\tilde{\mathcal{O}}(d^{-1/3})$. Our result is consistent with the rate obtained by Xu [41] and is faster than the rate $\mathcal{O}(d^{-1/4})$ obtained by Ganti et al [15].

**Example 5** (Structured tensors with repeating entries). We consider a more general model than that in Section 1. Consider an $r$-sign representable tensor $\Theta \in \mathscr{P}_{\text{sgn}}(r)$ with at most $d$ distinct entries with repetition pattern. Applying our results with $\alpha = \infty$ and $|\mathcal{N}| = d$ yields the rate $\tilde{\mathcal{O}}(d^{-(K-2)/2})$.

The following corollary reveal the sample complexity for nonparamtric tensor completion.

**Corollary 1** (Sample complexity for nonparametric completion). Assume the same conditions of Theorem 3 and bounded $|\mathcal{N}|$. Then, with high probability over $\mathcal{Y}_\Omega$,

$$\text{MAE}(\hat{\Theta}, \Theta) \to 0, \quad \text{as} \quad \frac{|\Omega|}{dr \log^2 |\Omega|} \to \infty.$$

Our result improves earlier work [42, 18, 31] by allowing both low- and high-rank signals. Interestingly, the sample requirements depend only on the sign complexity $dr$ but not the nonparametric complexity $\alpha$. Note that $\tilde{\mathcal{O}}(dr)$ roughly matches the degree of freedom of sign tensors, suggesting the optimality of our sample requirements.

## 4.2 Implementation via learning reduction

This section addresses the practical implementation of our estimation (7). We take a learning reduction approach by dividing the full procedure into a meta algorithm and $2H + 1$ base algorithms. The meta algorithm takes the average of $(2H + 1) \asymp \text{poly}(d)$ sign tensors, whereas each base algorithm estimates the tensor $\text{sgn}(\Theta - \pi)$ given binary input $\text{sgn}(\mathcal{Y} - \pi)$ and a target rank $r$. The full procedure is described in Algorithm 1 and Figure 2.

---

**Algorithm 1** Nonparametric tensor completion via learning reduction

---

**Input:** Noisy and incomplete data tensor $\mathcal{Y}_\Omega$, rank $r$, resolution parameter $H$, ridge penalty $\lambda$.
  1: **for** $\pi \in \mathcal{H} = \{-1, \ldots, -\frac{1}{H}, 0, \frac{1}{H}, \ldots, 1\}$ **do**
  2:   Define $\pi$-shifted tensor $\bar{\mathcal{Y}} = \mathcal{Y} - \pi$ and corresponding sign tensor $\text{sgn}(\bar{\mathcal{Y}}) = \text{sgn}(\mathcal{Y} - \pi)$.
  3:   Perform 1-bit tensor estimation algorithm [17, 39, 25, 2] on $\bar{\mathcal{Y}}_\Omega$ and obtain $\hat{\mathcal{Z}}_\pi \leftarrow \arg\min_{\text{low-rank } \mathcal{Z}} \sum_{\omega \in \Omega} |\bar{\mathcal{Y}}(\omega)| F(\mathcal{Z}(\omega)\text{sgn}\bar{\mathcal{Y}}(\omega)) + \lambda \|\mathcal{Z}\|_F^2$ where $F(\cdot)$ is the large-margin loss.
  4: **end for**
**Output:** Estimated signal tensor $\hat{\Theta}_F = \frac{1}{2H+1} \sum_{\pi \in \mathcal{H}} \text{sgn}(\hat{\mathcal{Z}}_\pi)$.

---

The base algorithm reduces to a low-rank 1-bit tensor estimation problem. Following the common practice in classification [6], we replace the 0-1 loss $\ell(z, y) = |\text{sgn}z - \text{sgn}y|$ in (4) with a continuous large-margin loss $F(m)$ where $m = z\text{sgn}(y)$ is the margin. Examples of large-margin loss are hinge loss $F(m) = (1-m)_+$, logistic loss $F(m) = \log(1+e^{-m})$, and $\psi$-loss $F(m) = 2\min(1, (1-m)_+)$ with $m_+ = \max(m, 0)$. A number of polynomial-time algorithms with convergence guarantees are readily available for this problem [17, 39, 25, 2]. We implement hinge loss [2, 16, 22] which maintains desirable statistical properties as in 0-1 loss, because of the linear excess risk bound [37]

$$\text{Risk}(\mathcal{Z}) - \text{Risk}(\Theta - \pi) \leq C[\text{Risk}_F(\mathcal{Z}) - \text{Risk}_F(\Theta - \pi)], \quad \text{for all } \pi \in [-1, 1] \text{ and all tensor } \mathcal{Z}.$$

Here $\text{Risk}_F(\cdot)$ is defined similarly as in (5) with hinge loss in place of 0-1 loss. The resulting estimate enjoys both statistical and computational efficiency under mild conditions.

**Theorem 4** (Large-margin loss). *Let $\hat{\Theta}_F$ be the output from Algorithm 1 with $F$ being the hinge loss. Under the set-up of Theorem 2 and technical assumptions on base algorithms, $\hat{\Theta}_F$ has the same error bound as in (9). Furthermore, the total complexity is within a $\text{poly}(d)$ factor of base algorithms.*

The full statement of Theorem 4 can be found in Appendix A.5. Technical assumptions in the theorem depend on the chosen base algorithm. For example, a signal-to-noise ratio for base problem is needed for the polynomial complexity of algorithms [39, 20]. We remark that we did not attempt to propose a new tensor algorithm. Instead, we present a learning reduction by adopting existing algorithms for a more challenging high-rank problems almost for free, i.e., at only an extra $\text{poly}(d)$ computational cost, but at almost no extra statistical cost. The developed sign-representable tensor model unifies low-rank and high-rank tensors, thereby empowering exiting algorithms for broader implications.

In principle, uses can choose their own favorite large-margin losses, as long as the base algorithms are sample efficient. The comparison between various large-margin losses has been studied before [6]. Note that, instead of using $\hat{\mathcal{Z}}_\pi$ as in existing 1-bit tensor algorithms [17, 39], we use $\text{sgn}(\hat{\mathcal{Z}}_\pi)$ for more challenging nonparametric estimation. The sign aggregation brings the benefits of flexibility and accuracy over classical low-rank models.

## 5  Numerical experiments

**Finite-sample accuracy.** We compare our nonparametric tensor method (**NonparaT**) with two alternative approaches: low-rank tensor CP decomposition (**CPT**), and the matrix version of our method applied to tensor unfolding (**NonparaM**). The performance is assessed under both complete and incomplete observations. We generate signal tensors based on four models summarized in Table 2, including block tensors, transformed low rank tensors, and structured tensors with repeating entries. We consider order-3 tensors of equal dimension, and set $d \in \{15, 20, \ldots, 55, 60\}$, $r \in \{2, 3, \ldots, 10\}$, $H = 10 + (d - 15)/5$ in Algorithm 1. All summary statistics are averaged across 30 replicates.

Table 2: Simulation models used for comparison. Here $\boldsymbol{M}_k \in \{0, 1\}^{d \times 3}$ denotes membership matrix; $\mathcal{C} \in \mathbb{R}^{3 \times 3 \times 3}$ is the block mean tensor; $\boldsymbol{a} = d^{-1}(1, 2, \ldots, d)^T$ is a length-$d$ vector; $\mathcal{Z}_{\max}$ and $\mathcal{Z}_{\min}$ are order-3 tensors with entries $d^{-1}\max(i, j, k)$ and $d^{-1}\min(i, j, k)$, respectively.

| Simulation | Signal Tensor $\Theta$ | Rank | Sign Rank | $\alpha$ | $|\mathcal{N}|$ | CDF of Tensor Entries | Noise |
|---|---|---|---|---|---|---|---|
| 1 | $\mathcal{C} \times \boldsymbol{M}_1 \times \boldsymbol{M}_2 \times \boldsymbol{M}_3$ | $3^3$ | $\leq 3^3$ | $\infty$ | $\leq 3^3$ | | Uniform $[-0.3, 0.3]$ |
| 2 | $|\boldsymbol{a} \otimes \boldsymbol{1} \otimes \boldsymbol{1} - \boldsymbol{1} \otimes \boldsymbol{a} \otimes \boldsymbol{1}|$ | $d$ | $\leq 3$ | $1$ | $0$ | | Normal $\mathcal{N}(0, 0.15)$ |
| 3 | $\log(0.5 + \mathcal{Z}_{\max})$ | $\geq d$ | $2$ | $\infty$ | $d$ | | Uniform $[-0.1, 0.1]$ |
| 4 | $2.5 - \exp(\mathcal{Z}_{\min}^{1/3})$ | $\geq d$ | $2$ | $\infty$ | $d$ | | Normal $\mathcal{N}(0, 0.15)$ |

Figure 3a-b compares the estimation error under full observation. For space consideration, only results for models 2-3 are presented in the main paper, and the rest in the Appendix C.1. We find that MAE decreases with tensor dimension for all three methods. Our method **NonparaT** achieves the best performance in all scenarios, whereas the second best method is **CPT** for model 2, and **NonparaM** for model 3. The model 2 has controlled multilinear rank along mode 3, which makes tensor methods **NonparaT** and **CPT** more accurate than **NonparaM**. The model 3 fits poorly into low-rank tensor families, and therefore, the two nonparametric methods **NonparaT** and **NonparaM** exhibit the greater advantage. Figure 3c-d shows the completion error against observation fraction. We fix $d = 40$ and gradually increase the observation fraction $|\Omega|/d^3$ from 0.3 to 1. Again, we find that **NonparaT** achieves the lowest error. The simulation covers a wide range of complexities, and our method shows good performance in experiments.

**Data applications.** We apply our method to two tensor datasets, the MRN-114 human brain connectivity data [38], and NIPS data [19]. The brain dataset records the structural connectivity among 68 brain regions for 114 individuals along with their Intelligence Quotient (IQ) scores. We organize the connectivity data into an order-3 tensor, where entries encode the presence or absence of fiber connections between brain regions across individuals. The NIPS dataset consists of word occurrence counts in papers published from 1987 to 2003. We focus on the top 100 authors, 200 most frequent words, and normalize each word count by log transformation with pseudo-count 1. The resulting dataset is an order-3 tensor with entry representing the log counts of words by authors across years.

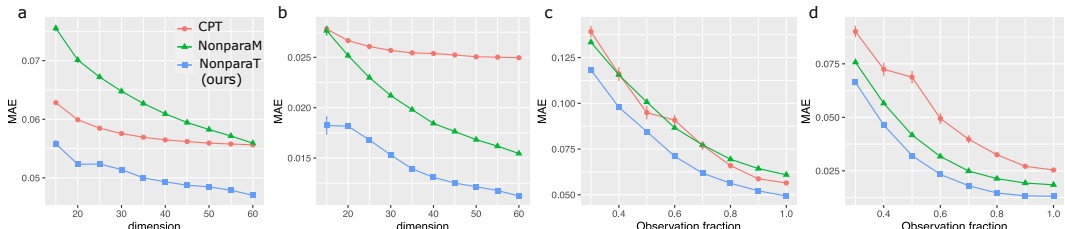

Figure 3: Performance comparison between different methods. (a)-(b): Estimation error versus tensor dimension. (c)-(d): Estimation error versus observation fraction. Panels (a) and (c) are for model 2, whereas (b) and (d) are for model 3.

Table 3 compares the prediction accuracy of different methods. Reported MAEs are averaged over five runs of cross-validation, with 20% entries for testing and 80% for training. Our method substantially outperforms the low-rank CP method for every configuration under consideration. Further increment of rank appears to have little effect on the performance, and we find that increased missingness gives more advantages to our method (see details in Appendix B.2). The comparison highlights the advantage of our method in achieving accuracy while maintaining low complexity.

Table 3: MAE comparison between **NonparaT** ($H = 20$) and **CPT** in the real data analysis. Standard errors are in parenthesis.

| MRN-114 brain connectivity dataset | | | |
|---|---|---|---|
| Method | $r = 6$ | $r = 9$ | $r = 12$ |
| **NonparaT (Ours)** | **0.14**(0.001) | **0.12**(0.001) | **0.12**(0.001) |
| CPT | 0.23(0.006) | 0.22(0.004) | 0.21(0.006) |
| NIPS word occurrence dataset | | | |
| Method | $r = 6$ | $r = 9$ | $r = 12$ |
| **NonparaT (Ours)** | **0.16**(0.002) | **0.15**(0.001) | **0.14**(0.001) |
| CPT | 0.20(0.007) | 0.19(0.007) | 0.17(0.007) |

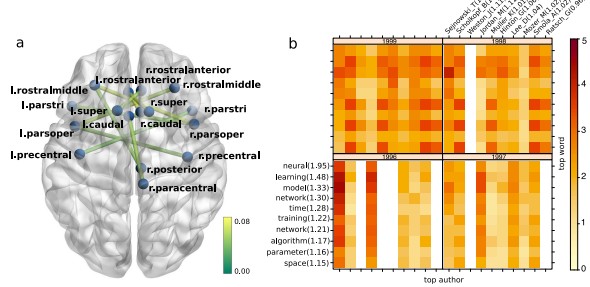

Figure 4: (a) top IQ-associated edges in the brain connectivity data. (b) top (authors, words, year) triplets in the NIPS data.

Figure 4a shows the top 10 brain edges based on regression analysis of denoised tensor from **NonparaT** against normalized IQ scores. We find that the top connections are mostly inter-hemisphere edges, consistent with recent research on brain connectivity [32, 38]. Figure 4b illustrates the results from NIPS data, where we plot the entries in $\hat{\Theta}$ corresponding to top authors and most-frequent words (after excluding generic words such as *figure*, *results*, etc). The identified pattern agrees with active topics in the NIPS publication. Among the top words are *neural* (marginal mean = 1.95), *learning* (1.48), and *network* (1.21), whereas top authors are *T. Sejnowski* (1.18), *B. Scholkopf* (1.17), *M. Jordan* (1.11), and *G. Hinton* (1.06). We also find strong heterogeneity among word occurrences across authors and years. For example, *training* and *algorithm* are popular words for *B. Scholkopf* and *A. Smola* in 1998-1999, whereas *model* occurs more often in *M. Jordan* and in 1996. The detected patterns and achieved accuracy demonstrate the applicability of our method.

## 6 Conclusion

We have developed a tensor estimation method that addresses both low- and high-rankness based on sign series representation. Our work provides a nonparametric framework for tensor estimation, and we establish accuracy guarantees for recovering a wide range of structured tensors. Our proposed learning reduction strategy empowers existing algorithms for broader implication, thereby connecting the low-rank (parametric) tensors and high-rank (nonparametric) tensors. We hope the work opens up new inquiry that allows more researchers to contribute to this field.

## Acknowledgements

This research is supported in part by NSF grant DMS- 1915978, NSF DMS-2023239, and Wisconsin Alumni Research Foundation.

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
