# Appendix for "Beyond the Signs: Nonparametric Tensor Completion via Sign Series"

Chanwoo Lee and Miaoyan Wang

Department of Statistics, University of Wisconsin - Madison

{chanwoo.lee, miaoyan.wang}@wisc.edu

The appendix consists of proofs (Section A), additional theoretical results (Section B), and numerical experiments (Section C).

## A Proofs

### A.1 Proofs of Propositions 1-2

*Proof of Proposition 1.*

Part (a). The strictly monotonicity of $g$ implies that the inverse function $g^{-1}\colon \mathbb{R} \to \mathbb{R}$ is well-defined. When $g$ is strictly increasing, the mapping $x \mapsto g(x)$ is sign preserving. Specifically, if $x \geq 0$, then $g(x) \geq g(0) = 0$. Conversely, if $g(x) \geq 0 = g(0)$, then applying $g^{-1}$ to both sides gives $x \geq 0$. When $g$ is strictly decreasing, the mapping $x \mapsto g(x)$ is sign reversing. Specifically, if $x \geq 0$, then $g(x) \leq g(0) = 0$. Conversely, if $g(x) \geq 0 = g(0)$, then applying $g^{-1}$ to both sides gives $x \leq 0$. Therefore, $\Theta \simeq g(\Theta)$, or $\Theta \simeq -g(\Theta)$. Since constant multiplication does not change the tensor rank, we have $\mathrm{srank}(\Theta) = \mathrm{srank}(g(\Theta)) \leq \mathrm{rank}(g(\Theta))$.

Part (b). See Section B.2 for constructive examples.

$\square$

*Proof of Proposition 2.* Fix $\pi \in [-1, 1]$. Based on the definition of classification loss $L(\cdot, \cdot)$, the function $\mathrm{Risk}(\cdot)$ relies only on the sign pattern of the tensor. Therefore, without loss of generality, we assume both $\bar{\Theta}, \mathcal{Z} \in \{-1, 1\}^{d_1 \times \cdots \times d_K}$ are binary tensors. We evaluate the excess risk

$$\mathrm{Risk}(\mathcal{Z}) - \mathrm{Risk}(\bar{\Theta}) = \mathbb{E}_{\omega \sim \Pi} \underbrace{\mathbb{E}_{\mathcal{Y}(\omega)} \left\{ |\mathcal{Y}(\omega) - \pi| \left[ \left| \mathcal{Z}(\omega) - \mathrm{sgn}(\bar{\mathcal{Y}}(\omega)) \right| - \left| \bar{\Theta}(\omega) - \mathrm{sgn}(\bar{\mathcal{Y}}(\omega)) \right| \right] \right\}}_{\overset{\mathrm{def}}{=} I(\omega)}. \quad (1)$$

Denote $y = \mathcal{Y}(\omega)$, $z = \mathcal{Z}(\omega)$, $\bar{\theta} = \bar{\Theta}(\omega)$, and $\theta = \Theta(\omega)$. The expression of $I(\omega)$ is simplified as

$$\begin{aligned}
I(\omega) &= \mathbb{E}_{y|\omega} \left[ (y - \pi)(\bar{\theta} - z)\mathbb{1}(y \geq \pi) + (\pi - y)(z - \bar{\theta})\mathbb{1}(y < \pi) \right] \\
&= \mathbb{E}_{y|\omega} \left[ (\bar{\theta} - z)(y - \pi) \right] \\
&= \left[ \mathrm{sgn}(\theta - \pi) - z \right](\theta - \pi) \\
&= |\mathrm{sgn}(\theta - \pi) - z||\theta - \pi| \geq 0, \quad (2)
\end{aligned}$$

where the third line uses the fact $\mathbb{E}y = \theta$ and $\bar{\theta} = \mathrm{sgn}(\theta - \pi)$, and the last line uses the assumption $z \in \{-1, 1\}$. The equality (2) is attained when $z = \mathrm{sgn}(\theta - \pi)$ or $\theta = \pi$. Combining (2) with (1), we conclude that, for all $\mathcal{Z} \in \{-1, 1\}^{d_1 \times \cdots \times d_K}$,

$$\mathrm{Risk}(\mathcal{Z}) - \mathrm{Risk}(\bar{\Theta}) = \mathbb{E}_{\omega \sim \Pi} |\mathrm{sgn}(\Theta(\omega) - \pi) - \mathcal{Z}(\omega)||\Theta(\omega) - \pi| \geq 0. \quad (3)$$

In particular, setting $\mathcal{Z} = \bar{\Theta} = \mathrm{sgn}(\Theta - \pi)$ in (3) yields the minimum. Therefore,

$$\mathrm{Risk}(\bar{\Theta}) = \min\{\mathrm{Risk}(\mathcal{Z})\colon \mathcal{Z} \in \mathbb{R}^{d_1 \times \cdots \times d_K}\} \leq \min\{\mathrm{Risk}(\mathcal{Z})\colon \mathrm{rank}(\mathcal{Z}) \leq r\}.$$

Since $\mathrm{srank}(\Theta - \pi) \leq r$ by assumption, the last inequality becomes equality. The proof is complete. $\qquad\square$

## A.2 Proof of Theorem 1

*Proof of Theorem 1.* Fix $\pi \notin \mathcal{N}$. Based on (3) in Proposition 2, we have

$$\mathrm{Risk}(\mathcal{Z}) - \mathrm{Risk}(\bar{\Theta}) = \mathbb{E}\left[\,|\mathrm{sgn}\mathcal{Z} - \mathrm{sgn}\bar{\Theta}||\bar{\Theta}|\,\right]. \tag{4}$$

The Assumption 1 states that

$$\mathbb{P}\left(|\bar{\Theta}| \leq t\right) \leq \begin{cases} ct^{\alpha}, & \text{for all } \Delta s \leq t < \rho(\pi, \mathcal{N}), \\ C\Delta s, & \text{for all } 0 \leq t < \Delta s. \end{cases} \tag{5}$$

Without further specification, all relevant probability statements, such as $\mathbb{E}$ and $\mathbb{P}$, are with respect to $\omega \sim \Pi$.

We divide the proof into two cases: $\alpha > 0$ and $\alpha = \infty$.

- Case 1: $\alpha > 0$.

By (4), for all $0 \leq t < \rho(\pi, \mathcal{N})$,

$$\begin{aligned}
\mathrm{Risk}(\mathcal{Z}) - \mathrm{Risk}(\bar{\Theta}) &\geq t\mathbb{E}\left(|\mathrm{sgn}\mathcal{Z} - \mathrm{sgn}\bar{\Theta}|\mathbb{1}\{|\bar{\Theta}| > t\}\right) \\
&\geq 2t\mathbb{P}\left(\mathrm{sgn}\mathcal{Z} \neq \mathrm{sgn}\bar{\Theta} \text{ and } |\bar{\Theta}| > t\right) \\
&\geq 2t\left\{\mathbb{P}\left(\mathrm{sgn}\mathcal{Z} \neq \mathrm{sgn}\bar{\Theta}\right) - \mathbb{P}\left(|\bar{\Theta}| \leq t\right)\right\} \\
&\geq t\left\{\mathrm{MAE}(\mathrm{sgn}\mathcal{Z}, \mathrm{sgn}\bar{\Theta}) - C\Delta s - 2ct^{\alpha}\right\}, \tag{6}
\end{aligned}$$

where the last line follows from the definition of MAE and (5). We maximize the lower bound (6) with respect to $t$, and obtain the optimal $t_{\mathrm{opt}}$,

$$t_{\mathrm{opt}} = \begin{cases} \rho(\pi, \mathcal{N}), & \text{if } \mathrm{MAE}(\mathrm{sgn}\mathcal{Z}, \mathrm{sgn}\bar{\Theta}) > \text{cut-off}, \\ \left[\frac{1}{2c(1+\alpha)}(\mathrm{MAE}(\mathrm{sgn}\mathcal{Z}, \mathrm{sgn}\bar{\Theta}) - C\Delta S)\right]^{1/\alpha}, & \text{if } \mathrm{MAE}(\mathrm{sgn}\mathcal{Z}, \mathrm{sgn}\bar{\Theta}) \leq \text{cut-off}. \end{cases}$$

where we have denoted the cut-off $= 2c(1+\alpha)\rho^{\alpha}(\pi, \mathcal{N}) + C\Delta s$. The corresponding lower bound of the inequality (6) becomes

$$\mathrm{Risk}(\mathcal{Z}) - \mathrm{Risk}(\bar{\Theta}) \geq \begin{cases} c_1\rho(\pi, \mathcal{N})\left[\mathrm{MAE}(\mathrm{sgn}\mathcal{Z}, \mathrm{sgn}\bar{\Theta}) - C\Delta s\right], & \text{if } \mathrm{MAE}(\mathrm{sgn}\mathcal{Z}, \mathrm{sgn}\bar{\Theta}) > \text{cut-off}, \\ c_2\left[\mathrm{MAE}(\mathrm{sgn}\mathcal{Z}, \mathrm{sgn}\bar{\Theta}) - C\Delta s\right]^{\frac{1+\alpha}{\alpha}}, & \text{if } \mathrm{MAE}(\mathrm{sgn}\mathcal{Z}, \mathrm{sgn}\bar{\Theta}) \leq \text{cut-off}, \end{cases}$$

where $c_1, c_2 > 0$ are two constants independent of $\mathcal{Z}$. Combining both cases gives

$$\begin{aligned}
\mathrm{MAE}(\mathrm{sgn}\mathcal{Z}, \mathrm{sgn}\bar{\Theta}) &\lesssim [\mathrm{Risk}(\mathcal{Z}) - \mathrm{Risk}(\bar{\Theta})]^{\frac{\alpha}{1+\alpha}} + \frac{1}{\rho(\pi, \mathcal{N})}\left[\mathrm{Risk}(\mathcal{Z}) - \mathrm{Risk}(\bar{\Theta})\right] + \Delta s \\
&\leq C(\pi)[\mathrm{Risk}(\mathcal{Z}) - \mathrm{Risk}(\bar{\Theta})]^{\frac{\alpha}{1+\alpha}} + \Delta s,
\end{aligned}$$

where $C(\pi) > 0$ is a multiplicative factor independent of $\mathcal{Z}$.

- Case 2: $\alpha = \infty$. The inequality (6) now becomes

$$\text{Risk}(\mathcal{Z}) - \text{Risk}(\bar{\Theta}) \geq t \left[ \text{MAE}(\text{sgn}\bar{\Theta}, \text{sgn}\mathcal{Z}) - C\Delta s \right], \quad \text{for all } 0 \leq t < \rho(\pi, \mathcal{N}). \tag{7}$$

The conclusion follows by taking $t = \frac{\rho(\pi, \mathcal{N})}{2}$ in the inequality (7).

$\square$

**Remark A.1.** The proof of Theorem 1 shows that, under global $\alpha$-smoothness of $\Theta$,

$$\text{MAE}(\text{sgn}\mathcal{Z}, \text{sgn}\bar{\Theta}) \lesssim [\text{Risk}(\mathcal{Z}) - \text{Risk}(\bar{\Theta})]^{\frac{\alpha}{1+\alpha}} + \frac{1}{\rho(\pi, \mathcal{N})} \left[ \text{Risk}(\mathcal{Z}) - \text{Risk}(\bar{\Theta}) \right] + \Delta s, \tag{8}$$

for all $\mathcal{Z} \in \mathbb{R}^{d_1 \times \cdots \times d_K}$. For fixed $\pi$, the second term is absorbed into the first term.

## A.3    Proof of Theorem 2

The following lemma provides the variance-to-mean relationship implied by the $\alpha$-smoothness of $\Theta$. The relationship plays a key role in determining the convergence rate based on empirical process theory (Shen and Wong, 1994); also see Theorem A.1.

**Lemma A.1** (Variance-to-mean relationship)**.** *Consider the same setup as in Theorem 2. Fix $\pi \notin \mathcal{N}$. Let $L(\mathcal{Z}, \bar{Y}_\Omega)$ be the $\pi$-weighted classification loss*

$$L(\mathcal{Z}, \bar{\mathcal{Y}}_\Omega) = \frac{1}{|\Omega|} \sum_{\omega \in \Omega} \underbrace{|\bar{\mathcal{Y}}(\omega)|}_{weight} \times \underbrace{|\text{sgn}\mathcal{Z}(\omega) - \text{sgn}\bar{\mathcal{Y}}(\omega)|}_{classification\ loss}$$

$$= \frac{1}{|\Omega|} \sum_{\omega \in \Omega} \ell_\omega(\mathcal{Z}, \bar{\mathcal{Y}}), \tag{9}$$

*where we have denoted the function $\ell_\omega(\mathcal{Z}, \bar{\mathcal{Y}}) \stackrel{def}{=} |\bar{\mathcal{Y}}(\omega)||\text{sgn}\mathcal{Z}(\omega) - \text{sgn}\bar{\mathcal{Y}}(\omega)|$. Under Assumption 1 of the $\alpha$-smoothness of $\Theta$, we have*

$$\text{Var}[\ell_\omega(\mathcal{Z}, \bar{\mathcal{Y}}_\Omega) - \ell_\omega(\bar{\Theta}, \bar{\mathcal{Y}}_\Omega)] \lesssim [\text{Risk}(\mathcal{Z}) - \text{Risk}(\bar{\Theta})]^{\frac{\alpha}{1+\alpha}} + \frac{1}{\rho(\pi, \mathcal{N})}[\text{Risk}(\mathcal{Z}) - \text{Risk}(\bar{\Theta})] + \Delta s, \tag{10}$$

*for all tensors $\mathcal{Z} \in \mathbb{R}^{d_1 \times \cdots \times d_K}$. Here the expectation and variance are taken with respect to both $\mathcal{Y}$ and $\omega \sim \Pi$.*

*Proof of Lemma A.1.* We expand the variance by

$$\text{Var}[\ell_\omega(\mathcal{Z}, \bar{\mathcal{Y}}_\Omega) - \ell_\omega(\bar{\Theta}, \bar{\mathcal{Y}}_\Omega)] \lesssim \mathbb{E}|\ell_\omega(\mathcal{Z}, \bar{\mathcal{Y}}_\Omega) - \ell_\omega(\bar{\Theta}, \bar{\mathcal{Y}}_\Omega)|^2$$
$$\lesssim \mathbb{E}|\ell_\omega(\mathcal{Z}, \bar{\mathcal{Y}}_\Omega) - \ell_\omega(\bar{\Theta}, \bar{\mathcal{Y}}_\Omega)|$$
$$\leq \mathbb{E}|\text{sgn}\mathcal{Z} - \text{sgn}\bar{\Theta}| = \text{MAE}(\text{sgn}\mathcal{Z}, \text{sgn}\bar{\Theta}), \tag{11}$$

where the second line comes from the boundedness of classification loss $L(\cdot, \cdot)$, and the third line comes from the inequality $||a - b| - |c - b|| \leq |a - b|$ for $a, b, c \in \{-1, 1\}$, together with the boundedness of classification weight $|\bar{\mathcal{Y}}(\omega)|$. Here we have absorbed the constant multipliers in $\lesssim$. The conclusion (10) then directly follows by applying Remark A.1 to (11). $\square$

*Proof of Theorem 2.* Fix $\pi \notin \mathcal{N}$. For notational simplicity, we suppress the subscript $\pi$ and write $\hat{\mathcal{Z}}$ in place of $\hat{\mathcal{Z}}_\pi$. Denote $n = |\Omega|$ and $\rho = \rho(\pi, \mathcal{N})$.

Because the classification loss $L(\cdot, \cdot)$ is scale-free, i.e., $L(\mathcal{Z}, \cdot) = L(c\mathcal{Z}, \cdot)$ for every $c > 0$, we consider the estimation subject to $\|\mathcal{Z}\|_F \leq 1$ without loss of generality. Specifically, let

$$\hat{\mathcal{Z}} = \underset{\mathcal{Z}: \, \text{rank}(\mathcal{Z}) \leq r, \|\mathcal{Z}\|_F \leq 1}{\arg\min} L(\mathcal{Z}, \bar{\mathcal{Y}}_\Omega). \tag{12}$$

We next apply the empirical process theory to bound $\hat{\mathcal{Z}}$. To facilitate the analysis, we view the data $\bar{\mathcal{Y}}_\Omega = \{\bar{\mathcal{Y}}(\omega): \omega \in \Omega\}$ as a collection of $n$ independent random variables where the randomness is from both $\bar{\mathcal{Y}}$ and $\omega \sim \Pi$. Write the index set $\Omega = \{1, \ldots, n\}$, so the loss function (9) becomes

$$L(\mathcal{Z}, \bar{\mathcal{Y}}_\Omega) = \frac{1}{n} \sum_{i=1}^n \ell_i(\mathcal{Z}, \bar{\mathcal{Y}}).$$

We use $f_{\mathcal{Z}}: [d_1] \times \cdots \times [d_n] \to \mathbb{R}$ to denote the function induced by tensor $\mathcal{Z}$ such that $f_{\mathcal{Z}}(\omega) = \mathcal{Z}(\omega)$ for $\omega \in [d_1] \times \cdots \times [d_K]$. Under this set-up, the quantity of interest

$$L(\mathcal{Z}, \bar{\mathcal{Y}}_\Omega) - L(\bar{\Theta}, \bar{\mathcal{Y}}_\Omega) = \frac{1}{n} \sum_{i=1}^n \underbrace{\left[ \ell_i(\mathcal{Z}, \bar{\mathcal{Y}}) - \ell_i(\bar{\Theta}, \bar{\mathcal{Y}}) \right]}_{\overset{\text{def}}{=} \Delta_i(f_{\mathcal{Z}}, \bar{\Theta})},$$

is an empirical process induced by function $f_{\mathcal{Z}} \in \mathcal{F}_{\mathcal{T}}$ where $\mathcal{T} = \{\mathcal{Z}: \text{rank}(\mathcal{Z}) \leq r, \ \|\mathcal{Z}\|_F \leq 1\}$. Note that there is an one-to-one correspondence between sets $\mathcal{F}_{\mathcal{T}}$ and $\mathcal{T}$.

Let $L_n$ denote the desired convergence rate to seek. By definition of $\hat{\mathcal{Z}}$ in (12), we have,

$$L(\hat{\mathcal{Z}}, \bar{\mathcal{Y}}_\Omega) - L(\bar{\Theta}, \bar{\mathcal{Y}}_\Omega) = \frac{1}{n} \sum_{i=1}^n \Delta_i(f_{\mathcal{Z}}, \bar{\Theta}) \leq 0.$$

Therefore, we have the following inclusion of probability events,

$$\left\{ (\omega, \mathcal{Y}_\omega): \text{Risk}(\hat{\mathcal{Z}}) - \text{Risk}(\bar{\Theta}) \geq L_n \right\}$$

$$\subset \left\{ (\omega, \mathcal{Y}_\omega): \exists \mathcal{Z} \text{ s.t. } \text{rank}(\mathcal{Z}) \leq r, \text{Risk}(\mathcal{Z}) - \text{Risk}(\bar{\Theta}) \geq L_n, \text{ and } \frac{1}{n} \sum_{i=1}^n \Delta_i(f_{\mathcal{Z}}, \bar{\Theta}) \leq 0 \right\}$$

$$\subset \left\{ (\omega, \mathcal{Y}_\omega): \sup_{\substack{\text{rank}(\mathcal{Z}) \leq r \\ \text{Risk}(\mathcal{Z}) - \text{Risk}(\bar{\Theta}) \geq L_n}} -\frac{1}{n} \sum_{i=1}^n \Delta_i(f_{\mathcal{Z}}, \bar{\Theta}) \geq 0 \right\}$$

$$\subset \bigcup_{\ell=1}^\infty \left\{ (\omega, \mathcal{Y}_\omega): \sup_{\mathcal{Z} \in A_\ell} -\frac{1}{n} \sum_{i=1}^n \Delta_i(f_{\mathcal{Z}}, \bar{\Theta}) \geq 0 \right\}, \tag{13}$$

where we have partitioned $\{\mathcal{Z}: \text{rank}(\mathcal{Z}) \leq r \text{ and } \text{Risk}(\mathcal{Z}) - \text{Risk}(\bar{\Theta}) \geq L_n\}$ in to union of $A_\ell$ with

$$A_\ell = \{\mathcal{Z}: \text{rank}(\mathcal{Z}) \leq r \text{ and } \ell L_n \leq \text{Risk}(\mathcal{Z}) - \text{Risk}(\bar{\Theta}) < (\ell+1)L_n\},$$

for $\ell = 1, 2, \ldots$. Let $\Gamma$ denote the target probability for the first line in (13). To bound $\Gamma$, we bound the sum of probability over the sets $A_\ell$. For each $A_\ell$, we consider the centered empirical process,

$$v_n(f_{\mathcal{Z}}) := -\frac{1}{n} \sum_{i=1}^n \left( \Delta_i(f_{\mathcal{Z}}, \bar{\Theta}) - \mathbb{E}\Delta_i(f_{\mathcal{Z}}, \bar{\Theta}) \right). \tag{14}$$

Notice $(\ell + 1)L_n \geq \mathbb{E}\Delta_i(f_{\mathcal{Z}}, \bar{\Theta}) = \mathrm{Risk}(\mathcal{Z}) - \mathrm{Risk}(\bar{\Theta}) \geq \ell L_n$ for all $\mathcal{Z} \in A_\ell$. Combining (13), (14) and union bound yields

$$\Gamma \leq \sum_{\ell=1}^{\infty} \mathbb{P}\left\{\sup_{\mathcal{Z} \in A_\ell} v_n(f_{\mathcal{Z}}) \geq \ell L_n =: M(\ell)\right\}. \tag{15}$$

Notice that, based on Lemma A.1, the variance of empirical process is bounded by

$$\sup_{\mathcal{Z} \in A_\ell} \mathrm{Var}\Delta_i(f_{\mathcal{Z}}, \bar{\Theta}) \lesssim \sup_{\mathcal{Z} \in A_\ell} \left(\left[\mathbb{E}\Delta_i(f_{\mathcal{Z}}, \bar{\Theta})\right]^{\frac{\alpha}{1+\alpha}} + \frac{1}{\rho}\mathbb{E}\Delta_i(f_{\mathcal{Z}}, \bar{\Theta})\right) + \Delta s$$

$$\leq M(\ell+1)^{\frac{\alpha}{1+\alpha}} + \frac{1}{\rho}M(\ell+1) + \Delta s =: V(\ell).$$

We next bound the right-hand side of (15) by choosing $L_n$ that satisfies conditions in Theorem A.1 (The specification of $L_n$ is deferred to the next paragraph). One such $L_n$ is chosen, Theorem A.1 gives us

$$\Gamma \lesssim \sum_{\ell=1}^{\infty} \exp\left(-\frac{nM^2(\ell)}{V(\ell) + 2M(\ell)}\right) \tag{16}$$

$$\lesssim \sum_{\ell=1}^{\infty} \exp(-\rho\ell n L_n)$$

$$\leq \left(\frac{e^{-n\rho L_n}}{1 - e^{-n\rho L_n}}\right).$$

Now, we specify $L_n$ that satisfies the condition of Theorem A.1. The quantity $L_n$ is determined by the solution to the following inequality,

$$\sup_{\ell \geq 1} \frac{1}{x} \int_x^{\sqrt{x^{\alpha/(\alpha+1)}+x/\rho+\Delta s}} \sqrt{\mathcal{H}_{[\,]}(\varepsilon, \mathcal{F}_{\mathcal{T}}, \|\cdot\|_2)} d\varepsilon \lesssim n^{1/2}, \quad \text{where } x = \ell L_n. \tag{17}$$

In particular, the smallest $L_n$ satisfying (17) yields the best upper bound of the error rate. Here $\mathcal{H}_{[\,]}(\varepsilon, \mathcal{F}_{\mathcal{T}}, \|\cdot\|_2)$ denotes the $L_2$-norm, $\varepsilon$-bracketing number (c.f. Definition A.1) for function family $\mathcal{F}_{\mathcal{T}}$.

Based on Lemma A.2, the inequality (17) is satisfied with the choice

$$L_n \asymp t_n^{(\alpha+1)/(\alpha+2)} + \frac{t_n}{\rho}, \quad \text{where } t_n = \left(\frac{d_{\max}rK\log n}{n}\right) \text{ and } d_{\max} := \max_{k \in [K]} d_k.$$

Finally, it follows from Theorem A.1 and (16) that

$$\mathbb{P}\left\{\mathrm{Risk}(\hat{\mathcal{Z}}) - \mathrm{Risk}(\bar{\Theta}) \geq L_n\right\} \lesssim \left(\frac{e^{-n\rho L_n}}{1 - e^{-n\rho L_n}}\right)$$

$$\lesssim e^{-nt_n},$$

where the last inequality uses the fact that $\rho L_n \gtrsim t_n \gtrsim \frac{1}{n}$ by our choice of $L_n$ and $t_n$.

Inserting the above bound into (8) gives that, with high probability at least $1 - \exp(-nt_n)$,

$$\mathrm{MAE}(\mathrm{sgn}\hat{\mathcal{Z}}, \mathrm{sgn}\bar{\Theta}) \lesssim [\mathrm{Risk}(\hat{\mathcal{Z}}) - \mathrm{Risk}(\bar{\Theta})]^{\alpha/(\alpha+1)} + \frac{1}{\rho}[\mathrm{Risk}(\hat{\mathcal{Z}}) - \mathrm{Risk}(\bar{\Theta})] + \Delta s$$

$$\lesssim t_n^{\alpha/(\alpha+2)} + \frac{1}{\rho^{\alpha/\alpha+1}}t_n^{\alpha/(\alpha+1)} + \frac{1}{\rho}t_n^{(\alpha+1)/(\alpha+2)} + \frac{1}{\rho^2}t_n$$

$$\leq 4t_n^{\alpha/(\alpha+2)} + \frac{4}{\rho^2}t_n, \tag{18}$$

where the second line uses the fact that $\Delta s \ll t_n$, and the last line follows from the fact that $a(b^2 + b^{(\alpha+2)/(\alpha+1)} + b + 1) \le 4a(b^2 + 1)$ with $a = \frac{t_n}{\rho^2}$ and $b = \rho t_n^{-1/(\alpha+2)}$. We plug $t_n$ into (18) and absorb the term $K$ into the constant. The conclusion is then proved by noting $n = |\Omega|$ by definition. $\qquad\square$

**Definition A.1** (Bracketing number). Consider a family of functions $\mathcal{F}$, and let $\varepsilon > 0$. Let $\mathcal{X}$ denote the domain space equipped with measure $\Pi$. We call $\{(f_m^l, f_m^u)\}_{m=1}^M$ an $L_2$-metric, $\varepsilon$-bracketing function set of $\mathcal{F}$, if for every $f \in \mathcal{F}$, there exists an $m \in [M]$ such that

$$f_m^l(x) \le f(x) \le f_m^u(x), \quad \text{for all } x \in \mathcal{X},$$

and

$$\|f_m^l - f_m^u\|_2 \stackrel{\text{def}}{=} \sqrt{\mathbb{E}_{x \sim \Pi} |f_m^l(x) - f_m^u(x)|^2} \le \varepsilon, \text{ for all } m = 1, \ldots, M.$$

The bracketing number with $L_2$-metric, denoted $\mathcal{H}_{[\ ]}(\varepsilon, \mathcal{F}, \|\cdot\|_2)$, is the logarithm of the smallest cardinality of the $\varepsilon$-bracketing function set of $\mathcal{F}$.

**Lemma A.2** (Bracketing complexity of low-rank tensors). *Define the family of rank-$r$ bounded tensors $\mathcal{T} = \{\mathcal{Z} \in \mathbb{R}^{d_1 \times \cdots \times d_K} : \text{rank}(\mathcal{Z}) \le r, \|\mathcal{Z}\|_F \le 1\}$ and the induced function family $\mathcal{F}_\mathcal{T} = \{f_\mathcal{Z} : \mathcal{Z} \in \mathcal{T}\}$. Set*

$$L_n \asymp \left(\frac{d_{\max} rK \log n}{n}\right)^{(\alpha+1)/(\alpha+2)} + \frac{1}{\rho(\pi, \mathcal{N})}\left(\frac{d_{\max} rK \log n}{n}\right), \text{ where } d_{\max} = \max_{k \in [K]} d_k.$$

*Then, the following inequality is satisfied provided that $\Delta s \lesssim n^{-1}$,*

$$\sup_{\ell \ge 1} \frac{1}{\ell L_n} \int_{\ell L_n}^{\sqrt{\ell L_n^{\alpha/(\alpha+1)} + \frac{\ell L_n}{\rho(\pi, \mathcal{N})} + \Delta s}} \sqrt{\mathcal{H}_{[\ ]}(\varepsilon, \mathcal{F}_\mathcal{T}, \|\cdot\|_2)} d\varepsilon \le Cn^{1/2}, \tag{19}$$

*where $C > 0$ is a constant independent of $r, K$ and $d_{max}$.*

*Proof of Lemma A.2.* To simplify the notation, we denote $\rho = \rho(\pi, \mathcal{N})$. Notice that

$$\|f_{\mathcal{Z}_1} - f_{\mathcal{Z}_1}\|_2 \le \|f_{\mathcal{Z}_1} - f_{\mathcal{Z}_1}\|_\infty \le \|\mathcal{Z}_1 - \mathcal{Z}_1\|_F \quad \text{for all } \mathcal{Z}_1, \mathcal{Z}_2 \in \mathcal{T}.$$

It follows from Kosorok (2007, Theorem 9.22) that the $L_2$-metric, $(2\epsilon)$-bracketing number of $\mathcal{F}_\mathcal{T}$ is bounded by

$$\mathcal{H}_{[\ ]}(2\varepsilon, \mathcal{F}_\mathcal{T}, \|\cdot\|_2) \le \mathcal{H}(\varepsilon, \mathcal{T}, \|\cdot\|_F) \le Cd_{\max} rK \log \frac{K}{\varepsilon}.$$

The last inequality is from the covering number bounds for rank-$r$ bounded tensors; see Mu et al. (2014, Lemma 3). Inserting the bracketing number into (19) gives

$$g(L, \ell) = \frac{1}{\ell L} \int_{\ell L}^{\sqrt{\ell L^{\alpha/(\alpha+1)} + \rho^{-1}\ell L + \Delta s}} \sqrt{d_{\max} rK \log\left(\frac{K}{\varepsilon}\right)} d\varepsilon. \tag{20}$$

Define $g(L) := \sup_{\ell \ge 1} g(L, \ell)$. By the monotonicity the integrand in (20), we bound $g(L)$ by

$$g(L) \le \sup_{\ell \ge 1} \frac{\sqrt{d_{\max} rK}}{\ell L} \int_{\ell L}^{\sqrt{\ell L^{\alpha/(\alpha+1)} + \rho^{-1}\ell L + n^{-1}}} \sqrt{\log\left(\frac{K}{\ell L}\right)} d\varepsilon$$

$$\le \sup_{\ell \ge 1} \sqrt{d_{\max} rK \log\left(\frac{K}{\ell L}\right)} \left(\frac{(\ell L)^{\alpha/(2\alpha+2)} + \sqrt{\rho^{-1}\ell L + n^{-1}}}{\ell L} - 1\right)$$

$$\lesssim \sqrt{d_{\max} rK \log(1/L)} \left[\frac{1}{L^{(\alpha+2)/(2\alpha+2)}} + \frac{1}{\sqrt{\rho L}}\left(1 + \frac{\rho}{2nL}\right)\right], \tag{21}$$

where the the second line follows from $\sqrt{a+b} \leq \sqrt{a} + \sqrt{b}$ for $a, b > 0$ and the last line comes from the fact that the bound achieves maximum when $\ell = 1$. It remains to verify that $g(L_n) \leq Cn^{1/2}$ for $L_n$ specified in (19). Plugging $L_n$ into the last line of (21) gives

$$g(L_n) \leq \sqrt{d_{\max} rK \log(1/L_n)} \left( \frac{1}{L_n^{(\alpha+2)/(2\alpha+2)}} + \frac{2}{\sqrt{\rho L_n}} \right)$$

$$\leq \sqrt{d_{\max} rK \log n} \left( \left[ \left( \frac{d_{\max} rK \log n}{n} \right)^{\frac{\alpha+1}{\alpha+2}} \right]^{-\frac{\alpha+2}{2\alpha+2}} + \left[ 2\rho \left( \frac{d_{\max} rK \log n}{\rho n} \right) \right]^{-\frac{1}{2}} \right)$$

$$\leq Cn^{1/2},$$

where $C > 0$ is a constant independent of $r, K$ and $d_{\max}$. The proof is therefore complete. $\square$

**Theorem A.1** (Theorem 3 in Shen and Wong (1994))**.** *Let $\mathcal{F}$ be a class of functions defined on $\mathcal{X}$ with $\sup_{f \in \mathcal{F}} \|f\|_\infty \leq T$. Let $(\boldsymbol{X}_i)_{i=1}^n$ be i.i.d. random variables with distribution $\mathbb{P}_{\boldsymbol{X}}$ over $\mathcal{X}$. Set $\sup_{f \in \mathcal{F}} \mathrm{Var} f(\boldsymbol{X}) = V < \infty$. Define the empirical process $\hat{\mathbb{E}} f = \frac{1}{n} \sum_{i=1}^n f(\boldsymbol{X}_i)$. Define $x_n^*$ to be the solution to the following inequality*

$$\frac{1}{x} \int_x^{\sqrt{V}} \sqrt{\mathcal{H}_{[\,]}(\varepsilon, \mathcal{F}, \|\cdot\|_2)} d\varepsilon \lesssim \sqrt{n}.$$

*Suppose $\sqrt{V} \leq T$ and*

$$x_n^* \lesssim \frac{V}{T}, \quad \text{and} \quad \mathcal{H}_{[\,]}(\sqrt{V}, \mathcal{F}, \|\cdot\|_2) \lesssim \frac{n(x_n^*)^2}{V}.$$

*Then, we have*

$$\mathbb{P}\left( \sup_{f \in \mathcal{F}} \hat{\mathbb{E}} f - \mathbb{E} f \geq x_n^* \right) \lesssim \exp\left( -\frac{n(x_n^*)^2}{V + Tx_n^*} \right).$$

### A.4 Proof of Theorem 3

*Proof of Theorem 3.* By definition of $\hat{\Theta}$, we have

$$\mathrm{MAE}(\hat{\Theta}, \Theta) = \mathbb{E} \left| \frac{1}{2H+1} \sum_{\pi \in \mathcal{H}} \mathrm{sgn} \hat{Z}_\pi - \Theta \right|$$

$$\leq \mathbb{E} \left| \frac{1}{2H+1} \sum_{\pi \in \mathcal{H}} \left( \mathrm{sgn} \hat{Z}_\pi - \mathrm{sgn}(\Theta - \pi) \right) \right| + \mathbb{E} \left| \frac{1}{2H+1} \sum_{\pi \in \mathcal{H}} \mathrm{sgn}(\Theta - \pi) - \Theta \right|$$

$$\leq \frac{1}{2H+1} \sum_{\pi \in \mathcal{H}} \mathrm{MAE}(\mathrm{sgn} \hat{Z}_\pi, \mathrm{sgn}(\Theta - \pi)) + \frac{1}{H}, \tag{22}$$

where the last line comes from the triangle inequality and the inequality

$$\left| \frac{1}{2H+1} \sum_{\pi \in \mathcal{H}} \mathrm{sgn}(\Theta(\omega) - \pi) - \Theta(\omega) \right| \leq \frac{1}{H}, \quad \text{for all } \omega \in [d_1] \times \cdots \times [d_K].$$

Write $n = |\Omega|$. Now it suffices to bound the first term in (22). For any given $t \geq t_n = \frac{d_{\max} rK \log n}{n}$, define the event

$$A = \left\{ \mathrm{MAE}(\mathrm{sgn} \hat{Z}_\pi, \mathrm{sgn}(\Theta - \pi)) \lesssim t^{\alpha/(2+\alpha)} + \frac{t}{\rho^2(\pi, \mathcal{N})} \text{ for all } \pi \in \mathcal{H} \right\}.$$

We shall prove that under the event $A$,

$$\frac{1}{2H+1} \sum_{\pi \in \mathcal{H}} \text{MAE}(\text{sgn}\hat{Z}_\pi, \text{sgn}(\Theta - \pi)) \lesssim t^{\alpha/(\alpha+2)} + \frac{1+|\mathcal{N}|}{H} + Ht. \tag{23}$$

Theorem 2 implies that the sign estimation accuracy depends on the closeness of $\pi \in \mathcal{H}$ to the mass points in $\mathcal{N}$. Therefore, we partition the level set $\pi \in \mathcal{H}$ based on their closeness to $\mathcal{N}$. Specifically, Define $\mathcal{H}_1 \stackrel{\text{def}}{=} \{\pi \in \mathcal{H} \colon \rho(\pi, \mathcal{N}) < \frac{1}{H}\}$ and $\mathcal{H}_2 = \mathcal{H} \setminus \mathcal{H}_1$. Notice $|\mathcal{H}_1| \leq 2|\mathcal{N}|$. We expand the left hand side of (23) by

$$\frac{1}{2H+1} \sum_{\pi \in \mathcal{H}} \text{MAE}(\text{sgn}\hat{Z}_\pi, \text{sgn}(\Theta - \pi))$$

$$= \frac{1}{2H+1} \sum_{\pi \in \mathcal{H}_1} \text{MAE}(\text{sgn}\hat{Z}_\pi, \text{sgn}(\Theta - \pi)) + \frac{1}{2H+1} \sum_{\pi \in \mathcal{H}_2} \text{MAE}(\text{sgn}\hat{Z}_\pi, \text{sgn}(\Theta - \pi)). \tag{24}$$

The first term involves only $2|\mathcal{N}|$ many number of summnmands thus can be bounded by $4|\mathcal{N}|/(2H+1)$. We bound the second term using the explicit forms of $\rho(\pi, \mathcal{N})$ in the sequence $\pi \in \mathcal{H}_2$. Under the event $A$, we have

$$\frac{1}{2H+1} \sum_{\pi \in \mathcal{H}_2} \text{MAE}(\text{sgn}\hat{\mathcal{Z}}_\pi, \text{sgn}(\Theta - \pi)) \lesssim \frac{1}{2H+1} \sum_{\pi \in \mathcal{H}_2} t^{\alpha/(\alpha+2)} + \frac{t}{2H+1} \sum_{\pi \in \mathcal{H}_2} \frac{1}{\rho^2(\pi, \mathcal{N})}$$

$$\leq t^{\alpha/(\alpha+2)} + \frac{t}{2H+1} \sum_{\pi \in \mathcal{H}_2} \sum_{\pi' \in \mathcal{N}} \frac{1}{|\pi - \pi'|^2}$$

$$\leq t^{\alpha/(\alpha+2)} + \frac{t}{2H+1} \sum_{\pi' \in \mathcal{N}} \sum_{\pi \in \mathcal{H}_2} \frac{1}{|\pi - \pi'|^2}$$

$$\leq t^{\alpha/(\alpha+2)} + 2CHt,$$

where the first inequality uses the property of event $A$, and the last inequality follows from Lemma A.3. Combining the bounds for the two terms in (24) completes the proof for conclusion (23); that is

$$\mathbb{P}\left(\text{MAE}(\hat{\Theta}, \Theta) \lesssim t^{\alpha/(\alpha+2)} + \frac{1+|\mathcal{N}|}{H} + Ht\right) \geq \mathbb{P}(A). \tag{25}$$

Based on the proof of Theorem 2 and union bound over $\pi \in \mathcal{H}$, we have, for all $t \geq t_n$,

$$\mathbb{P}(A) \geq 1 - \sum_{\pi \in \mathcal{H}} \mathbb{P}\left(\text{MAE}(\text{sgn}\hat{\mathcal{Z}}_\pi, \text{sgn}(\Theta - \pi)) \gtrsim t^{\alpha/(\alpha+2)} + \frac{t}{\rho(\pi, \mathcal{N})^2}\right)$$

$$\gtrsim 1 - (2H+1)\exp(-nt) \gtrsim 1 - \exp(-nt + \log H). \tag{26}$$

We choose $t \asymp t_n \log H$ in (26) so that $\log H$ is negligible compared to $nt$. Finally, combining (25) and (26) with the choice of $t$ yields

$$\text{MAE}(\hat{\Theta}, \Theta) \lesssim \left(\frac{d_{\max}rK \log|\Omega| \log H}{|\Omega|}\right)^{\alpha/(\alpha+2)} + \frac{1+|\mathcal{N}|}{H} + \frac{d_{\max}rK \log|\Omega|}{|\Omega|} H \log H,$$

with at least probability $1 - \exp(-d_{\max}rK \log|\Omega| \log H) \geq 1 - \exp(-d_{\max}rK \log|\Omega|)$.

$\square$

**Lemma A.3.** *Fix $\pi' \in \mathcal{N}$ and a sequence $\Pi = \{-1, \ldots, -1/H, 0, 1/H, \ldots, 1\}$ with $H \geq 2$. Then,*

$$\sum_{\pi \in \mathcal{H}_2} \frac{1}{|\pi - \pi'|^2} \leq 4H^2.$$

*Proof of Lemma A.3.* Notice that all points $\pi \in \mathcal{H}_2$ satisfy $|\pi - \pi'| \gtrsim \frac{1}{H}$ for all $\pi' \in \mathcal{N}$ by definition and the fact that $\Delta s$ is negligible compared to $1/H$. We use this fact to compute the sum

$$
\begin{aligned}
\sum_{\pi \in \mathcal{H}_2} \frac{1}{|\pi - \pi'|^2} &= \sum_{\frac{h}{H} \in \mathcal{H}_2} \frac{1}{|\frac{h}{H} - \pi'|^2} \\
&\leq 2H^2 \sum_{h=1}^{H} \frac{1}{h^2} \\
&\leq 2H^2 \left\{ 1 + \int_1^2 \frac{1}{x^2} dx + \int_2^3 \frac{1}{x^2} dx + \cdots + \int_{H-1}^{H} \frac{1}{x^2} dx \right\} \\
&= 2H^2 \left( 1 + \int_1^H \frac{1}{x^2} dx \right) \leq 4H^2,
\end{aligned}
$$

where the third line uses the monotonicity of $\frac{1}{x^2}$ for $x \geq 1$. $\qquad\square$

## A.5  Formal statement and proof of Theorem 4

Write $\bar{\mathcal{Y}} = \mathcal{Y} - \pi$, $\bar{\Theta} = \Theta - \pi$, and $n = |\Omega|$. Here we consider the estimation

$$\hat{\mathcal{Z}}_\pi = \underset{\mathrm{rank}(\mathcal{Z}) \leq r}{\arg\min} \sum_{\omega \in \Omega} |\bar{\mathcal{Y}}(\omega)| \times F(\mathcal{Z}(\omega) \mathrm{sgn}(\bar{\mathcal{Y}}(\omega)) + \lambda \|\mathcal{Z}\|_F^2, \tag{27}$$

where $\lambda > 0$ is the penalty parameter and $F$ is a large-margin loss satisfying the following assumption.

**Assumption A.1** (Assumptions on surrogate loss)**.**

(a) *(Approximation error) For any given $\pi \in [-1, 1]$, assume there exist a sequence of tensors $\mathcal{Z}_\pi^{(n)} \in \mathscr{P}_{\mathrm{sgn}}(r)$, such that $\mathrm{Risk}_F(\mathcal{Z}_\pi^{(n)}) - \mathrm{Risk}_F(\bar{\Theta}) \leq a_n$, for some sequence $a_n \to 0$ as $n \to \infty$. Furthermore, assume $\|\mathcal{Z}_\pi^{(n)}\|_F \leq J$ for some constant $J > 0$.*

(b) $F(z) = (1 - z)_+$ *is hinge loss.*

Assumption A.1(a) quantifies the representation capability of and $\mathscr{P}_{\mathrm{sgn}}(r)$. Assumption A.1(b) implies the Fisher consistency bound for the weighted risk (Scott, 2011),

$$\mathrm{Risk}(\mathcal{Z}) - \mathrm{Risk}(\bar{\Theta}) \lesssim \mathrm{Risk}_F(\mathcal{Z}) - \mathrm{Risk}_F(\bar{\Theta}), \text{ for all } \pi \in [-1, 1] \text{ and all } \mathcal{Z}.$$

Therefore, it suffices to bound the excess $F$-risk in order to bound the usual 0-1 risk. Under Assumption A.1, we establish the estimation accuracy guarantee for the large-margin estimators (27).

**Theorem A.2** (Large-margin estimation)**.** *Consider the same setup as in Theorem 3, and denote $t_n = \frac{d_{\max} r K \log n}{n}$. Suppose the surrogate loss $F$ satisfies Assumption A.1 with $a_n \lesssim t_n^{(\alpha+1)/(\alpha+2)}$. Set $\lambda \asymp t_n^{(\alpha+1)/(\alpha+2)} + t_n/\rho(\pi, \mathcal{N})$ in (27). Then, with high probability at least $1 - \exp(-nt_n)$, we have:*

(a) (Sign tensor estimation). For all $\pi \in [-1, 1]$ except for a finite number of levels,

$$\text{MAE}(\text{sgn}(\hat{\mathcal{Z}}_\pi), \text{sgn}(\bar{\Theta})) \lesssim t_n^{\frac{\alpha}{2+\alpha}} + \frac{1}{\rho^2(\pi, \mathcal{N})} t_n. \tag{28}$$

(b) (Tensor estimation).

$$\text{MAE}(\hat{\Theta}, \Theta) \lesssim (t_n \log H)^{\frac{\alpha}{2+\alpha}} + \frac{1 + |\mathcal{N}|}{H} + t_n H \log H. \tag{29}$$

In particualr, setting $H \asymp (1 + |\mathcal{N}|)^{1/2} t_n^{-1/2}$ yields the tightest upper bound in (29).

*Proof of Theorem A.2.* The tensor estimation error (29) directly follows from sign tensor estimation error (28) and the proof of Theorem 3. Therefore, it suffices to prove (28). Our proof uses the same techniques used in the proof of Theorem 2. We summarize only the key difference.

Fix $\pi \notin \mathcal{N}$. For notational simplicity, we suppress the subscript $\pi$ and write $\hat{\mathcal{Z}}$ in place of $\hat{\mathcal{Z}}_\pi$. Denote $n = |\Omega|$ and $\rho = \rho(\pi, \mathcal{N})$. Define $\ell_{\omega, F}(\mathcal{Z}) = |\bar{\mathcal{Y}}(\omega)| \times F(\mathcal{Z}(\omega) \text{sgn}(\bar{\mathcal{Y}}(\omega))$ and $\ell_{\omega, F'}(\mathcal{Z}) = |\bar{\mathcal{Y}}(\omega)| \times F'(\mathcal{Z}(\omega) \text{sgn}(\bar{\mathcal{Y}}(\omega))$ where $F'$ is T-truncated version of $F$ such that $F'(x) = \min(F(x), T)$ with $T = \max(2, J^2)$. We focus on the following two empirical processes induced by function $f_{\mathcal{Z}} \in \mathcal{F}_\mathcal{T}$ where $\mathcal{T} = \{\mathcal{Z} : \text{rank}(\mathcal{Z}) \leq r\}$,

$$\frac{1}{n} \sum_{i=1}^{n} \underbrace{[\ell_{i,F}(\mathcal{Z}, \bar{\mathcal{Y}}) - \ell_{i,F}(\bar{\Theta}, \bar{\mathcal{Y}})]}_{\overset{\text{def}}{=} \Delta_{i,F}(f_{\mathcal{Z}}, \bar{\Theta})}, \quad \text{and} \quad \frac{1}{n} \sum_{i=1}^{n} \underbrace{[\ell_{i,F'}(\mathcal{Z}, \bar{\mathcal{Y}}) - \ell_{i,F'}(\bar{\Theta}, \bar{\mathcal{Y}})]}_{\overset{\text{def}}{=} \Delta_{i,F'}(f_{\mathcal{Z}}, \bar{\Theta})}.$$

Note that there is an one-to-one correspondence between sets $\mathcal{F}_\mathcal{T}$ and $\mathcal{T}$.

By definition of $\hat{\mathcal{Z}}$ in (27), we have

$$\frac{1}{n} \sum_{i=1}^{n} \Delta_{i,F}(f_{\hat{\mathcal{Z}}}, \mathcal{Z}^{(n)}) \leq \lambda J^2 - \lambda \|\hat{\mathcal{Z}}\|_F^2,$$

where $\mathcal{Z}^{(n)}$ is a sequence of function in Assumption A.1(a). Let $L_n$ denote the desired convergence rate to seek. Then, we have the following inclusion of probability events,

$$\left\{ (\omega, \mathcal{Y}_\omega) : \text{Risk}_{F'}(\hat{\mathcal{Z}}) - \text{Risk}_{F'}(\bar{\Theta}) \geq 2L_n \right\}$$

$$\subset \left\{ (\omega, \mathcal{Y}_\omega) : \exists \mathcal{Z} \text{ s.t. } \text{rank}(\mathcal{Z}) \leq r, \text{Risk}_{F'}(\mathcal{Z}) - \text{Risk}_{F'}(\bar{\Theta}) \geq 2L_n, \right.$$

$$\left. \text{and} \ -\frac{1}{n} \sum_{i=1}^{n} \Delta_{i,F}(f_{\mathcal{Z}}, \mathcal{Z}^{(n)}) + \lambda J^2 - \lambda \|\hat{\mathcal{Z}}\|_F^2 \geq 0 \right\}$$

$$\overset{(*)}{\subset} \left\{ (\omega, \mathcal{Y}_\omega) : \exists \mathcal{Z} \text{ s.t. } \text{rank}(\mathcal{Z}) \leq r, \text{Risk}_{F'}(\mathcal{Z}) - \text{Risk}_{F'}(\bar{\Theta}) \geq 2L_n, \right.$$

$$\left. \text{and} \ -\frac{1}{n} \sum_{i=1}^{n} \Delta_{i,F'}(f_{\mathcal{Z}}, \mathcal{Z}^{(n)}) + \lambda J^2 - \lambda \|\hat{\mathcal{Z}}\|_F^2 \geq 0 \right\}$$

$$\subset \left\{ (\omega, \mathcal{Y}_\omega) : \sup_{\substack{\text{rank}(\mathcal{Z}) \leq r \\ \text{Risk}_{F'}(\mathcal{Z}) - \text{Risk}_{F'}(\bar{\Theta}) \geq 2L_n}} -\frac{1}{n} \sum_{i=1}^{n} \Delta_{i,F'}(f_{\mathcal{Z}}, \mathcal{Z}^{(n)}) + \lambda J^2 - \lambda \|\hat{\mathcal{Z}}\|_F^2 \geq 0 \right\}$$

$$\subset \bigcup_{\ell_1, \ell_2=1}^{\infty} \left\{ (\omega, \mathcal{Y}_\omega) : \sup_{\mathcal{Z} \in A_{\ell_1, \ell_2}} -\frac{1}{n} \sum_{i=1}^{n} \Delta_{i,F'}(f_{\mathcal{Z}}, \mathcal{Z}^{(n)}) + \lambda J^2 - \lambda \|\hat{\mathcal{Z}}\|_F^2 \geq 0 \right\}, \tag{30}$$

where $(*)$ comes from the fact

$$\ell_{\omega,F'}(\mathcal{Z},\bar{\mathcal{Y}}) \leq \ell_{\omega,F}(\mathcal{Z},\bar{\mathcal{Y}}) \text{ for all } \mathcal{Z}, \quad \text{and } \ell_{\omega,F'}(\mathcal{Z}^{(n)},\bar{\mathcal{Y}}) = \ell_{\omega,F}(\mathcal{Z}^{(n)},\bar{\mathcal{Y}}),$$

because the truncation constant $T = \max(2, J^2) \geq \max(2, \sup_n \|\mathcal{Z}^{(n)}\|_F^2)$. In the last line of (30), we have partitioned $\{\mathcal{Z} \colon \text{rank}(\mathcal{Z}) \leq r \text{ and } \text{Risk}_{F'}(\mathcal{Z}) - \text{Risk}_{F'}(\bar{\Theta}) \geq 2L_n\}$ into union of $A_{\ell_1,\ell_2}$ with

$$A_{\ell_1,\ell_2} = \Big\{ \mathcal{Z} \colon \text{rank}(\mathcal{Z}) \leq r, (\ell_1 + 1)L_n \leq \text{Risk}_{F'}(\mathcal{Z}) - \text{Risk}_{F'}(\bar{\Theta}) < (\ell_1 + 2)L_n,$$

$$\text{and } (\ell_2 - 1)J^2 \leq \|\mathcal{Z}\|_F^2 < \ell_2 J^2 \Big\},$$

for $\ell_1, \ell_2 = 1, 2, \ldots$.

Let $\Gamma$ denote the target probability for the first line in (30). For each $A_{\ell_1,\ell_2}$, we consider the centered empirical process,

$$v_n(f_{\mathcal{Z}}) := -\frac{1}{n} \sum_{i=1}^{n} \Big( \Delta_{i,F'}(f_{\mathcal{Z}}, \mathcal{Z}^{(n)}) - \mathbb{E}\Delta_{i,F'}(f_{\mathcal{Z}}, \mathcal{Z}^{(n)}) \Big). \tag{31}$$

Notice that

$$\mathbb{E}\Delta_{i,F'}(f_{\mathcal{Z}}, \mathcal{Z}^{(n)}) = \text{Risk}_{F'}(\mathcal{Z}) - \text{Risk}_{F'}(\bar{\Theta}) + \text{Risk}_{F'}(\bar{\Theta}) - \text{Risk}_{F'}(\mathcal{Z}^{(n)})$$
$$\geq (\ell_1 + 1)L_n - a_n$$
$$\geq \ell_1 L_n,$$

where the first inequality is from the fact that $\mathcal{Z} \in A_{\ell_1,\ell_2}$ and Assumption A.1(a), and the last inequality uses the condition that $a_n \lesssim L_n$.

Combining (30), (31) and the union bound yields

$$\Gamma \leq \sum_{\ell_1,\ell_2=1}^{\infty} \mathbb{P}\left\{ \sup_{\mathcal{Z}\in A_{\ell_1,\ell_2}} v_n(f_{\mathcal{Z}}) \geq \ell_1 L_n + \lambda(\ell_2 - 2)J^2 =: M(\ell_1, \ell_2) \right\}. \tag{32}$$

Similar to the proof of Lemma A.1 and Lemma 2 with $T$-truncated hinge loss in Lee et al. (2021), the variance of empirical process is bounded by

$$\sup_{\mathcal{Z}\in A_{\ell_1,\ell_2}} \text{Var}\Delta_{i,F'}(f_{\mathcal{Z}}, \bar{\Theta}) \lesssim \sup_{\mathcal{Z}\in A_{\ell_1,\ell_2}} \left( [\mathbb{E}\Delta_{i,F'}(f_{\mathcal{Z}}, \bar{\Theta})]^{\frac{\alpha}{1+\alpha}} + \frac{1}{\rho}\mathbb{E}\Delta_{i,F'}(f_{\mathcal{Z}}, \bar{\Theta}) \right) + \Delta s$$

$$\lesssim M(\ell_1, \ell_2)^{\frac{\alpha}{1+\alpha}} + \frac{1}{\rho}M(\ell_1, \ell_2) + \Delta s =: V(\ell_1, \ell_2).$$

To apply Theorem A.1, we choose the pair $(L_n, \lambda)$ satisfying

$$\sup_{\ell_1,\ell_2\geq 1} \frac{1}{x} \int_x^{\sqrt{x^{\alpha/(\alpha+1)}+x/\rho+\Delta s}} \sqrt{\mathcal{H}_{[\,]}(\varepsilon, \mathcal{F}_{\mathcal{T}}(\ell_2), \|\cdot\|_2)} d\varepsilon \lesssim n^{1/2}, \tag{33}$$

where $x = \ell_1 L_n + \lambda(\ell_2 - 2)J^2$ and $\mathcal{F}_{\mathcal{T}}(\ell_2) := \{f_{\mathcal{Z}} \colon \text{rank}(\mathcal{Z}) \leq r, \|\mathcal{Z}\|_F^2 \leq \ell_2 J^2\}$. Similar to the proof of Lemma A.2, we solve the pair $(L_n, \lambda)$ satisfying (33) as

$$L_n \asymp t_n^{(\alpha+1)/(\alpha+2)} + \frac{t_n}{\rho}, \quad \text{and} \quad \lambda = \frac{L_n}{2J^2}, \tag{34}$$

where $t_n = \frac{d_{\max} rK \log n}{n}$. With the choice (34), we bound the right-hand side of (32) based on Theorem A.1,

$$\Gamma \lesssim \sum_{\ell_1,\ell_2=1}^{\infty} \exp\left(-\frac{nM^2(\ell_1,\ell_2)}{V(\ell_1,\ell_2) + 2M(\ell_1,\ell_2)}\right)$$

$$\lesssim \sum_{\ell_1,\ell_2=1}^{\infty} \exp(-\rho n M(\ell_1,\ell_2))$$

$$\leq \left(\frac{e^{-n\rho L_n}}{1 - e^{-n\rho L_n}}\right)\left(\frac{e^{n\rho\lambda J^2}}{1 - e^{-n\rho\lambda J^2}}\right)$$

$$\lesssim e^{-n\rho L_n} \leq e^{-n t_n},$$

where the last line uses the fact that $2\rho\lambda J^2 = \rho L_n \gtrsim t_n \gtrsim n^{-1}$ from (34). The proof is then completed by (18). $\qquad\square$

# B  Additional results

## B.1  Sensitivity of tensor rank to monotonic transformations

In Section 1 of the main paper, we have provided a motivating example to show the sensitivity of tensor rank to monotonic transformations. Here, we describe the details of the example set-up.

The step 1 is to generate a rank-3 tensor $\mathcal{Z}$ based on the CP representation

$$\mathcal{Z} = \boldsymbol{a}^{\otimes 3} + \boldsymbol{b}^{\otimes 3} + \boldsymbol{c}^{\otimes 3},$$

where $\boldsymbol{a}, \boldsymbol{b}, \boldsymbol{c} \in \mathbb{R}^{30}$ are vectors consisting of $N(0,1)$ entries, and the shorthand $\boldsymbol{a}^{\otimes 3} = \boldsymbol{a} \otimes \boldsymbol{a} \otimes \boldsymbol{a}$ denotes the Kronecker power. We then apply $f(z) = (1 + \exp(-cz))^{-1}$ to $\mathcal{Z}$ entrywise, and obtain a transformed tensor $\Theta = f(\mathcal{Z})$.

The step 2 is to determine the rank of $\Theta$. Unlike matrices, the exact rank determination for tensors is NP hard. Therefore, we choose to compute the numerical rank of $\Theta$ as an approximation. The numerical rank is determined as the minimal rank for which the relative approximation error is below 0.1, i.e.,

$$\hat{r}(\Theta) = \min\left\{s \in \mathbb{N}_+ : \min_{\hat{\Theta}:\, \mathrm{rank}(\hat{\Theta}) \leq s} \frac{\|\Theta - \hat{\Theta}\|_F}{\|\Theta\|_F} \leq 0.1\right\}.$$

We compute $\hat{r}(\Theta)$ by searching over $s \in \{1, \ldots, 30^2\}$, where for each $s$, we (approximately) solve the least-square minimization using built-in `cp` function in R package `rTensor` with default setting (iteration = 25, tolerance = $10^{-5}$). We repeat steps 1-2 ten times, and plot the averaged numerical rank of $\Theta$ versus transformation level $c$ in Figure 1a.

## B.2  Tensor rank and sign-rank

In the main paper, we have provided several tensor examples with high tensor rank but low sign-rank. This section provides more examples and their proofs. Unless otherwise specified, let $\Theta$ be an order-$K$ $(d, \ldots, d)$-dimensional tensor.

**Example B.1** (Structured tensors with repeating entries)**.** Suppose the tensor $\Theta$ takes the form

$$\Theta(i_1, \ldots, i_K) = \log\left(1 + \frac{1}{d}\max(i_1, \ldots, i_K)\right), \text{ for all } (i_1, \ldots, i_K) \in [d]^K.$$

Then
$$\text{rank}(\Theta) \geq d, \quad \text{and} \quad \text{srank}(\Theta - \pi) \leq 2 \text{ for all } \pi \in \mathbb{R}.$$

**Remark B.1** (Connection with hypergraphon models)**.** This example is related to hypergraphons (Zhao, 2015; Lovász and Szegedy, 2006). Hypergraphon is a limiting function based on a sequence of uniform hypergraphs in cut distance (Zhao, 2015). Though hypergraphon is an important application, the implication of our results should be interpreted with cautions for two reasons:

(i) Unlike the matrix case where graphon is represented as a bivariate function, general hyper-graphons for order- tensors should be represented as $(2^K - 2)$-variate function (Zhao, 2015, Section 1.2). Our example depends on $K$ coordinates only, and in this sense, our example shares the common ground as simple hypergraphons (Kallenberg, 1999).

(ii) Unlike typical simple hypergraphons where the design points are random variables $x_i \sim \text{Uniform}[0,1]$ , our example uses deterministic design points $x_i = i/d$ . These two choices lead to a notable difference in the RMSE rate $d^{-(K-1)/3}$ (ours) vs. $d^{-1}$ (simple hypergraphon) (Balasubramanian, 2021). This improvement stems from the distinction of fixed vs. random designs. Whether it is possible to extend our theory to general hypergraphon is an interesting question for future research.

*Proof of Example B.1.* We first prove the results for $K = 2$. The full-rankness of $\Theta$ is verified from elementary row operations as follows

$$\begin{pmatrix} (\Theta_2 - \Theta_1)/(\log(1+\frac{2}{d}) - \log(1+\frac{1}{d})) \\ (\Theta_3 - \Theta_2)/(\log(1+\frac{3}{d}) - \log(1+\frac{2}{d})) \\ \vdots \\ (\Theta_d - \Theta_{d-1})/(\log(1+\frac{d}{d}) - \log(1+\frac{d-1}{d})) \\ \Theta_d / \log(1+\frac{d}{d}) \end{pmatrix} = \begin{pmatrix} 1 & 0 & \ddots & \ddots & 0 \\ 1 & 1 & \ddots & \ddots & \ddots \\ \vdots & \vdots & \ddots & \ddots & \ddots \\ 1 & 1 & 1 & 1 & 0 \\ 1 & 1 & 1 & 1 & 1 \end{pmatrix},$$

where $\Theta_i$ denotes the $i$-th row of $\Theta$. Now it suffices to show $\text{srank}(\Theta - \pi) \leq 2$ for $\pi$ in the feasible range $(\log(1 + \frac{1}{d}), \log 2)$. In this case, there exists an index $i^* \in \{2, \ldots, d\}$, such that $\log(1 + \frac{i^*-1}{d}) < \pi \leq \log(1 + \frac{i^*}{d})$. By definition, the sign matrix $\text{sgn}(\Theta - \pi)$ takes the form

$$\text{sgn}(\Theta(i,j) - \pi) = \begin{cases} -1, & \text{both } i \text{ and } j \text{ are smaller than } i^*; \\ 1, & \text{otherwise.} \end{cases} \tag{35}$$

Therefore, the matrix $\text{sgn}(\Theta - \pi)$ is a rank-2 block matrix, which implies $\text{srank}(\Theta - \pi) = 2$.

We now extend the results to $K \geq 3$. By definition of the tensor rank, the rank of a tensor is lower bounded by the rank of its matrix slice. So we have $\text{rank}(\Theta) \geq \text{rank}(\Theta(:,:,1,\ldots,1)) = d$. For the sign rank with feasible $\pi$, notice that the sign tensor $\text{sgn}(\Theta - \pi)$ takes the similar form as in (35),

$$\text{sgn}(\Theta(i_1, \ldots, i_K) - \pi) = \begin{cases} -1, & i_k < i^* \text{ for all } k \in [K]; \\ 1, & \text{otherwise,} \end{cases} \tag{36}$$

where $i^*$ denotes the index that satisfies $\log(1 + \frac{i^*-1}{d}) < \pi \leq \log(1 + \frac{i^*}{d})$. The equation (36) implies that $\text{sgn}(\Theta - \pi) = -2a^{\otimes K} + 1$, where $a = (1, \ldots, 1, 0, \ldots, 0)^T$ takes 1 on the $i$-th entry if $i < i^*$ and 0 otherwise. Henceforth $\text{srank}(\Theta - \pi) = 2$. □

In fact, Example B.1 is a special case of the following proposition.

**Proposition B.1** (Structured tensors with repeating entries). *Let $g\colon \mathbb{R} \to \mathbb{R}$ be a continuous function such that $g(z) = 0$ has at most $r \geq 1$ distinct real roots. For given numbers $x_{i_k}^{(k)} \in [0,1]$ for all $i_k \in [d_k]$, define a tensor $\Theta \in \mathbb{R}^{d_1 \times \cdots \times d_K}$ with entries*

$$\Theta(i_1, \ldots, i_K) = g(\max(x_{i_1}^{(1)}, \ldots, x_{i_K}^{(K)})), \quad (i_1, \ldots, i_K) \in [d_1] \times \cdots \times [d_K]. \tag{37}$$

*Then, the sign rank of $(\Theta - \pi)$ satisfies*

$$\mathrm{srank}(\Theta - \pi) \leq 2r.$$

*The same conclusion holds if we use* $\min$ *in place of* $\max$ *in* (37).

*Proof of Proposition B.1.* We reorder the tensor indices along each mode such that $x_1^{(k)} \leq \cdots \leq x_{d_k}^{(k)}$ for all $k \in [K]$. Based on the construction of $\mathcal{Z}_{\max}$, the reordering does not change the rank of $\mathcal{Z}_{\max}$ or $(\Theta - \pi)$. Let $z_1 < \cdots < z_r$ be the $r$ distinct real roots for the equation $g(z) = \pi$. We separate the proof for two cases, $r = 1$ and $r \geq 2$.

- When $r = 1$. The continuity of $g(\cdot)$ implies that the function $(g(z) - \pi)$ has at most one sign change point. Using similar proof as in Example B.1, we have

$$\mathrm{sgn}(\Theta - \pi) = 1 - 2\boldsymbol{a}^{(1)} \otimes \cdots \otimes \boldsymbol{a}^{(K)} \quad \text{or} \quad \mathrm{sgn}(\Theta - \pi) = 2\boldsymbol{a}^{(1)} \otimes \cdots \otimes \boldsymbol{a}^{(K)} - 1,$$

  where $\boldsymbol{a}^{(k)}$ are binary vectors defined by

$$\boldsymbol{a}^{(k)} = (\ \underbrace{1, \ldots, 1,}_{\text{positions for which } x_{i_k}^k < z_1}\ 0, \ldots, 0)^T, \quad \text{for } k \in [K].$$

  Therefore, $\mathrm{srank}(\Theta - \pi) \leq \mathrm{rank}(\mathrm{sgn}(\Theta - \pi)) = 2$.
- When $r \geq 2$. By continuity, the function $(g(z) - \pi)$ is non-zero and remains an unchanged sign in each of the intervals $(z_s, z_{s+1})$ for $1 \leq s \leq r - 1$. Define the index set

$$\mathcal{I} = \{s \in \mathbb{N}_+ \colon \text{the interval } (z_s, z_{s+1}) \text{ in which } g(z) < \pi\}.$$

  We now prove that the sign tensor $\mathrm{sgn}(\Theta - \pi)$ has rank bounded by $2r - 1$. To see this, consider the tensor indices for which $\mathrm{sgn}(\Theta - \pi) = -1$,

$$
\begin{aligned}
\{\omega \colon \Theta(\omega) - \pi < 0\} &= \{\omega \colon g(\mathcal{Z}_{\max}(\omega)) < \pi\} \\
&= \cup_{s \in \mathcal{I}} \{\omega \colon \mathcal{Z}_{\max}(\omega) \in (z_s, z_{s+1})\} \\
&= \cup_{s \in \mathcal{I}} \Big( \{\omega \colon x_{i_k}^{(k)} < z_{s+1} \text{ for all } k \in [K]\} \cap \{\omega \colon x_{i_k}^{(k)} \leq z_s \text{ for all } k \in [K]\}^c \Big).
\end{aligned}
\tag{38}
$$

  The equation (38) is equivalent to

$$\mathbb{1}(\Theta(i_1, \ldots, i_K) < \pi) = \sum_{s \in \mathcal{I}} \left( \prod_k \mathbb{1}(x_{i_k}^{(k)} < z_{s+1}) - \prod_k \mathbb{1}(x_{i_k}^{(k)} \leq z_s) \right), \tag{39}$$

  for all $(i_1, \ldots, i_K) \in [d_1] \times \cdots \times [d_K]$, where $\mathbb{1}(\cdot) \in \{0, 1\}$ denotes the indicator function. The equation (39) implies the low-rank representation of $\mathrm{sgn}(\Theta - \pi)$,

$$\mathrm{sgn}(\Theta - \pi) = 1 - 2\sum_{s \in \mathcal{I}} \left( \boldsymbol{a}_{s+1}^{(1)} \otimes \cdots \otimes \boldsymbol{a}_{s+1}^{(K)} - \bar{\boldsymbol{a}}_s^{(1)} \otimes \cdots \otimes \bar{\boldsymbol{a}}_s^{(K)} \right), \tag{40}$$

where $\boldsymbol{a}_{s+1}^{(k)}, \boldsymbol{a}_s^{(k)}$ are binary vectors defined by

$$\boldsymbol{a}_{s+1}^{(k)} = (\ \underbrace{1,\ldots,1,}_{\text{positions for which } x_{i_k}^{(k)} < z_{s+1}}\quad 0,\ldots,0)^T, \quad \text{and} \quad \bar{\boldsymbol{a}}_s^{(k)} = (\ \underbrace{1,\ldots,1,}_{\text{positions for which } x_{i_k}^{(k)} \leq z_s}\quad 0,\ldots,0)^T.$$

Therefore, by (40) and the assumption $|\mathcal{I}| \leq r - 1$, we conclude that

$$\text{srank}(\Theta - \pi) \leq 1 + 2(r - 1) = 2r - 1.$$

Combining two cases yields that $\text{srank}(\Theta - \pi) \leq 2r$ for any $r \geq 1$. $\qquad\square$

We next provide several additional examples such that $\text{rank}(\Theta) \geq d$ whereas $\text{srank}(\Theta) \leq c$ for a constant $c$ independent of $d$. We state the examples in the matrix case, i.e, $K = 2$. Similar conclusion extends to $K \geq 3$, by the following proposition.

**Proposition B.2** (Rank relationship between matrices and tensors). *Let $\boldsymbol{M} \in \mathbb{R}^{d_1 \times d_2}$ be a matrix. For any given $K \geq 3$, define an order-$K$ tensor $\Theta \in \mathbb{R}^{d_1 \times \cdots \times d_K}$ by*

$$\Theta = \boldsymbol{M} \otimes \mathbf{1}_{d_3} \otimes \cdots \otimes \mathbf{1}_{d_K},$$

*where $\mathbf{1}_{d_k} \in \mathbb{R}^{d_k}$ denotes an all-one vector, for $3 \leq k \leq K$. Then we have*

$$\text{rank}(\Theta) = \text{rank}(\boldsymbol{M}), \quad and \quad \text{srank}(\Theta - \pi) = \text{srank}(\boldsymbol{M} - \pi) \text{ for all } \pi \in \mathbb{R}.$$

*Proof of Proposition B.2.* The conclusion directly follows from the definition of tensor rank. $\qquad\square$

**Example B.2** (Stacked banded matrices). Let $\boldsymbol{a} = (1, 2, \ldots, d)^T$ be a $d$-dimensional vector, and define a $d$-by-$d$ banded matrix $\boldsymbol{M} = |\boldsymbol{a} \otimes \mathbf{1} - \mathbf{1} \otimes \boldsymbol{a}|$. Then

$$\text{rank}(\boldsymbol{M}) = d, \quad \text{and} \quad \text{srank}(\boldsymbol{M} - \pi) \leq 3, \quad \text{for all } \pi \in \mathbb{R}.$$

*Proof of Example B.2.* Note that $\boldsymbol{M}$ is a banded matrix with entries

$$\boldsymbol{M}(i, j) = |i - j|, \quad \text{for all } (i, j) \in [d]^2.$$

Elementary row operation shows that $\boldsymbol{M}$ is full rank as follows,

$$\begin{pmatrix} (\boldsymbol{M}_1 + \boldsymbol{M}_d)/(d-1) \\ \boldsymbol{M}_1 - \boldsymbol{M}_2 \\ \boldsymbol{M}_2 - \boldsymbol{M}_3 \\ \vdots \\ \boldsymbol{M}_{d-1} - \boldsymbol{M}_d \end{pmatrix} = \begin{pmatrix} 1 & 1 & 1 & \cdots & 1 & 1 \\ -1 & 1 & 1 & \cdots & 1 & 1 \\ -1 & -1 & 1 & \cdots & 1 & 1 \\ \vdots & \vdots & \vdots & \vdots & \vdots & \vdots \\ -1 & -1 & -1 & \cdots & -1 & 1 \end{pmatrix}.$$

We now show $\text{srank}(\boldsymbol{M} - \pi) \leq 3$ by construction. Define two vectors $\boldsymbol{b} = (2^{-1}, 2^{-2}, \ldots, 2^{-d})^T \in \mathbb{R}^d$ and $\text{rev}(\boldsymbol{b}) = (2^{-d}, \ldots, 2^{-1})^T \in \mathbb{R}^d$. We construct the following matrix

$$\boldsymbol{A} = \boldsymbol{b} \otimes \text{rev}(\boldsymbol{b}) + \text{rev}(\boldsymbol{b}) \otimes \boldsymbol{b}. \tag{41}$$

The matrix $\boldsymbol{A} \in \mathbb{R}^{d \times d}$ is banded with entries

$$\boldsymbol{A}(i, j) = \boldsymbol{A}(j, i) = \boldsymbol{A}(d - i, d - j) = \boldsymbol{A}(d - j, d - i) = 2^{-d-1}\left(2^{j-i} + 2^{i-j}\right), \quad \text{for all } (i, j) \in [d]^2.$$

Furthermore, the entry value $\boldsymbol{A}(i, j)$ decreases with respect to $|i - j|$; i.e.,

$$\boldsymbol{A}(i, j) \geq \boldsymbol{A}(i', j'), \quad \text{for all } |i - j| \geq |i' - j'|. \tag{42}$$

Notice that for a given $\pi \in \mathbb{R}$, there exists $\pi' \in \mathbb{R}$ such that $\text{sgn}(\boldsymbol{A} - \pi') = \text{sgn}(\boldsymbol{M} - \pi)$. This is because both $\boldsymbol{A}$ and $\boldsymbol{M}$ are banded matrices satisfying monotonicity (42). By definition (41), $\boldsymbol{A}$ is a rank-2 matrix. Henceforce, $\text{srank}(\boldsymbol{M} - \pi) = \text{srank}(\boldsymbol{A} - \pi') \leq 3$. $\qquad\square$

**Remark B.2.** The tensor analogy of banded matrices $\Theta = |\boldsymbol{a} \otimes \mathbf{1} \otimes \mathbf{1} - \mathbf{1} \otimes \boldsymbol{a} \otimes \mathbf{1}|$ is used as simulation model 3 in the main paper.

**Example B.3** (Stacked identity matrices)**.** Let $\boldsymbol{I}$ be a $d$-by-$d$ identity matrix. Then

$$\text{rank}(\boldsymbol{I}) = d, \quad \text{and} \quad \text{srank}(\boldsymbol{I} - \pi) \leq 3 \text{ for all } \pi \in \mathbb{R}.$$

*Proof of Proposition B.3.* Depending on the value of $\pi$, the sign matrix $\text{sgn}(\boldsymbol{I} - \pi)$ falls into one of the two cases:

(a) $\text{sgn}(\boldsymbol{I} - \pi)$ is a matrix of all 1, or of all $-1$;

(b) $\text{sgn}(\boldsymbol{I} - \pi) = 2\boldsymbol{I} - \mathbf{1}_d \otimes \mathbf{1}_d$.

The first cases are trivial, so it suffices to show $\text{srank}(\boldsymbol{I} - \pi) \leq 3$ in the third case.
 Based on Example B.2, the rank-2 matrix $\boldsymbol{A}$ in (41) satisfies

$$\boldsymbol{A}(i, j) \begin{cases} = 2^{-d}, & i = j, \\ \geq 2^{-d} + 2^{-d-2}, & i \neq j. \end{cases}$$

Therefore, $\text{sgn}\left(2^{-d} + 2^{-d-3} - \boldsymbol{A}\right) = 2\boldsymbol{I} - \mathbf{1}_d \otimes \mathbf{1}_d$. We conclude that $\text{srank}(\boldsymbol{I} - \pi) \leq \text{rank}(2^{-d} + 2^{-d-3} - \boldsymbol{A}) = 3$. $\qquad\square$

## B.3 Extension of Theorems 2-3 to unbounded observation with sub-Gaussian noise

Consider the signal plus noise model

$$\mathcal{Y} = \Theta + \mathcal{E},$$

where $\mathcal{E}$ consists of zero-mean, independent noise entries, and $\Theta \in \mathscr{P}_{\text{sgn}}(r)$ is an $\alpha$-smooth tensor. Theoretical results in Section 4 of the main paper are based on bounded observation $\|\mathcal{Y}\|_\infty \leq 1$. We extend the results to unbounded observation with the following assumption.

**Assumption B.1** (Sub-Gaussian noise)**.**

1. *There exists a constant $\beta > 0$, independent of tensor dimension, such that $\|\Theta\|_\infty \leq \beta$. Without loss of generality, we set $\beta = 1$.*

2. *The noise entries $\mathcal{E}(\omega)$ are independent zero-mean sub-Gaussian random variables with variance proxy $\sigma^2 > 0$; i.e, $\mathbb{P}(|\mathcal{E}(\omega)| \geq B) \leq 2e^{-B^2/2\sigma^2}$ for all $B > 0$.*

We say that an event $A$ occurs "with high probability" if $\mathbb{P}(A)$ tends to 1 as the tensor dimension $d_{\min} = \min_k d_k \to \infty$. The following result show that the sub-Gaussian noise incurs an additional $\log |\Omega|$ factor compared to the bounded case.

**Theorem B.1** (Extension to sub-Gaussian noise). *Consider the same condition of Theorem 2. Suppose that Assumption B.1 holds. With high probability over training data $\mathcal{Y}_\Omega$, we have:*

(a) *(Sign matrix estimation). For all $\pi \notin \mathcal{N}$,*

$$\text{MAE}(\text{sgn}(\hat{\mathcal{Z}}_\pi), \text{sgn}(\Theta - \pi)) \lesssim t_d^{\frac{\alpha}{\alpha+2}} + \frac{t_d}{\rho^2(\pi, \mathcal{N})}, \quad \text{where } t_d := \frac{r\sigma^2 d_{\max} \log d_{\max} \log |\Omega|}{|\Omega|}.$$

(b) *For all resolution parameter $H \in \mathbb{N}_+$,*

$$\text{MAE}(\hat{\Theta}, \Theta) \lesssim (t_d \log H)^{\alpha/(\alpha+2)} + \frac{1 + |\mathcal{N}|}{H} + H(t_d \log H). \tag{43}$$

*In particular, setting $H \asymp \left(\frac{1+|\mathcal{N}|}{t_d}\right)^{1/2}$ yields the tightest upper bound in (43).*

*Proof of Theorem B.1.* By setting $s = K\log(d_{\max})$ in Lemma B.1, we have

$$\mathbb{P}(\|\mathcal{E}\|_\infty \geq \sqrt{4\sigma^2 K \log d_{\max}}) \leq 2d_{\max}^{-K}.$$

We divide the sample space into two exclusive events:

- Event I: $\|\mathcal{E}\|_\infty \geq \sqrt{4\sigma^2 K \log d_{\max}}$;
- Event II: $\|\mathcal{E}\|_\infty < \sqrt{4\sigma^2 K \log d_{\max}}$.

Because the Event I occurs with probability tending to zero, we restrict ourselves to the Event II only, by following the proof of Theorem 2. We summarize the key difference compared to Section A. We expand the variance by

$$\begin{aligned}
\text{Var}\left[\ell_\omega\left(\mathcal{Z}, \bar{\mathcal{Y}}_\Omega\right) - \ell_\omega\left(\bar{\Theta}, \bar{\mathcal{Y}}_\Omega\right)\right] &\leq \mathbb{E}|\ell_\omega(\mathcal{Z}(\omega), \bar{\mathcal{Y}}(\omega)) - \ell_\omega(\bar{\Theta}(\omega), \bar{\mathcal{Y}}(\omega))|^2 \\
&= \mathbb{E}|\bar{\mathcal{Y}}(\omega) - \bar{\Theta}(\omega) + \bar{\Theta}(\omega)|^2 |\text{sgn}\mathcal{Z}(\omega) - \text{sgn}\bar{\Theta}(\omega)| \\
&\leq 2\left(4\sigma^2 K \log d_{\max} + 2\right) \mathbb{E}|\text{sgn}\mathcal{Z} - \text{sgn}\bar{\Theta}| \\
&\lesssim (\sigma^2 K \log d_{\max})\text{MAE}(\text{sgn}\mathcal{Z}, \text{sgn}\bar{\Theta}), \tag{44}
\end{aligned}$$

where the third line uses the facts $\|\bar{\Theta}\|_\infty \leq 2$ and $\|\bar{\mathcal{Y}} - \bar{\Theta}\|_\infty^2 = \|\mathcal{E}\|_\infty^2 < 4\sigma^2 K \log d_{\max}$ within the Event II; the last line comes from the definition of MAE and the asymptotic $\sigma^2 \log d_{\max} \gg 1$ provided that $\sigma > 0$ with $d_{\max}$ sufficiently large.

Based on (44), the $\alpha$-smoothness of $\Theta$ implies that for all measurable functions $f_{\mathcal{Z}}$, we have

$$\text{Var}\Delta_i(f_{\mathcal{Z}}, \bar{\Theta}) \lesssim \left(\sigma^2 K \log d_{\max}\right) \left\{ \left[\mathbb{E}\Delta_i(f_{\mathcal{Z}}, \bar{\Theta})\right]^{\frac{\alpha}{1+\alpha}} + \frac{1}{\rho}\mathbb{E}\Delta_i(f_{\mathcal{Z}}, \bar{\Theta}) + \Delta s \right\}. \tag{45}$$

Based on the proof of Theorem 2, the empirical process with variance-to-mean relationship (45) gives that

$$\mathbb{P}\left(\text{Risk}(\hat{\mathcal{Z}}) - \text{Risk}(\bar{\Theta}) \geq L_n\right) \lesssim \exp(-nt_n), \tag{46}$$

where the convergence rate $L_n$ is obtained by the same way in the proof of Lemma A.2,

$$L_n \asymp t_n^{(\alpha+1)/(\alpha+2)} + \frac{1}{\rho}t_n, \quad \text{with } t_n = \frac{r\sigma^2 d_{\max} \log d_{\max} \log n}{n}, \tag{47}$$

where constants (possibly depending on $K$) have been absorbed into the $\asymp$ relationship. Combining (46) and (47), we obtain that, with high probability,

$$\text{Risk}(\hat{\mathcal{Z}}) - \text{Risk}(\bar{\Theta}) \lesssim \left( \frac{r\sigma^2 d_{\max} \log d_{\max} \log |\Omega|}{|\Omega|} \right)^{(\alpha+1)/(\alpha+2)} + \frac{1}{\rho(\pi, \mathcal{N})} \left( \frac{r\sigma^2 d_{\max} \log d_{\max} \log |\Omega|}{|\Omega|} \right),$$
(48)

Therefore, combining (48) and (18) completes the proof. The tensor estimation error follows readily from the proof of Theorem 3 and Theorem B.1. □

**Lemma B.1** (sub-Gaussian maximum). *Let $X_1, \ldots, X_n$ be independent sub-Gaussian zero-mean random variables with variance proxy $\sigma^2$. Then, for any $s > 0$,*

$$\mathbb{P}\left\{ \max_{1 \leq i \leq n} |X_i| \geq \sqrt{2\sigma^2(\log n + s)} \right\} \leq 2e^{-s}.$$

*Proof of Lemma B.1.* The conclusion follows from

$$\mathbb{P}[\max_{1 \leq i \leq n} |X_i| \geq u] \leq \sum_{i=1}^{n} \mathbb{P}[X_i \geq u] \leq 2ne^{-\frac{u^2}{2\sigma^2}} = 2e^{-s},$$

where we set $u = \sqrt{2\sigma^2(\log n + s)}$. □

# C   Additional results on numerical experiments

## C.1   Simulations

Section 5 of the main paper has summarized the major findings. Here we provide more detailed simulation results for models 1-4.

Figure S1 compares the estimation error under full observation for models 1-4. Similar to results for models 2-3 in the main paper, we find that the MAE decreases with tensor dimension for all three methods. Our method **NonParaT** achieves the best performance in all scenarios, whereas the second best method is **CPT** for models 1-2, and **NonParaM** for models 3-4. As explained in the main paper, models 1-2 have controlled multilinear tensor rank, which makes tensor methods **NonParaT** and **CPT** more accurate than matrix methods. For models 3-4, the rank exceeds the tensor dimension, and therefore, the two nonparametric methods **NonParaT** and **NonparaM** exhibit the greater advantage for signal recovery.

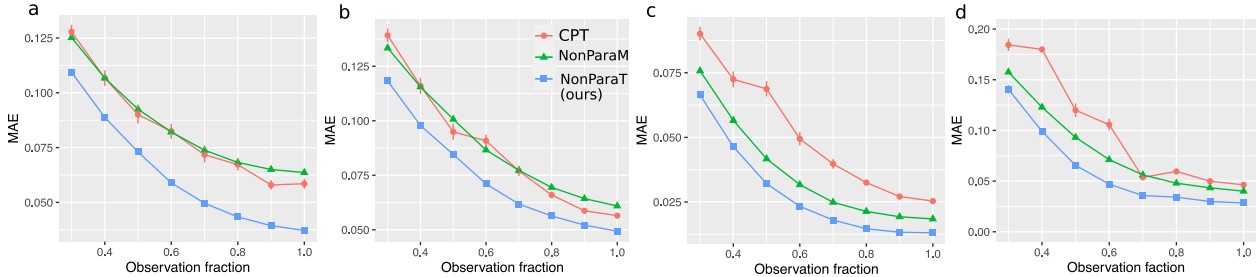

Supplementary Figure S2: Completion error versus observation fraction. Panels (a)-(d) correspond to simulation models 1-4 in Table 2.

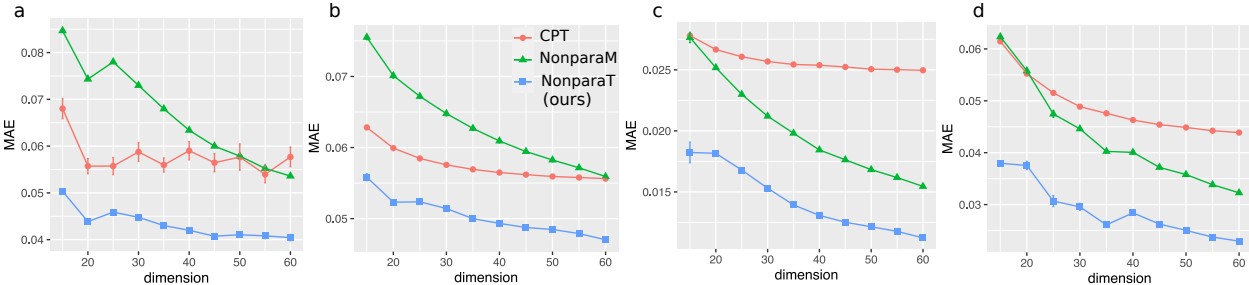

Supplementary Figure S1: Estimation error versus tensor dimension. Panels (a)-(d) correspond to simulation models 1-4 in Table 2.

Figure S2 shows the completion error against observation fraction. We find that **NonParaT** achieves the lowest error among all methods. Our simulation covers a reasonable range of complexities; for example, model 1 has $3^3$ jumps in the CDF of signal $\Theta$, and models 2 and 4 have unbounded noise. Nevertheless, our method shows good performance in spite of model misspecification. This robustness is appealing in practice because the structure of underlying signal tensor is often unknown.

## C.2 Brain connectivity analysis

Figure S3 shows the MAE based on 5-fold cross-validations with $r = 3, 6, \ldots, 15$ and $H = 20$. We find that our method outperforms CPT in all combinations of ranks and missing rates. The achieved error reduction appears to be more profound as the missing rate increases. This trend highlights the applicability of our method in tensor completion tasks. In addition, our method exhibits a smaller standard error in cross-validation experiments as shown in Figure S3 and Table 3 (in the main paper), demonstrating the stability over CPT. One possible reason is that that our estimate is guaranteed to be in $[0, 1]$ (for binary tensor problem where $\mathcal{Y} \in \{0, 1\}^{d_1 \times \cdots \times d_K}$) whereas CPT estimation may fall outside the valid range $[0, 1]$.

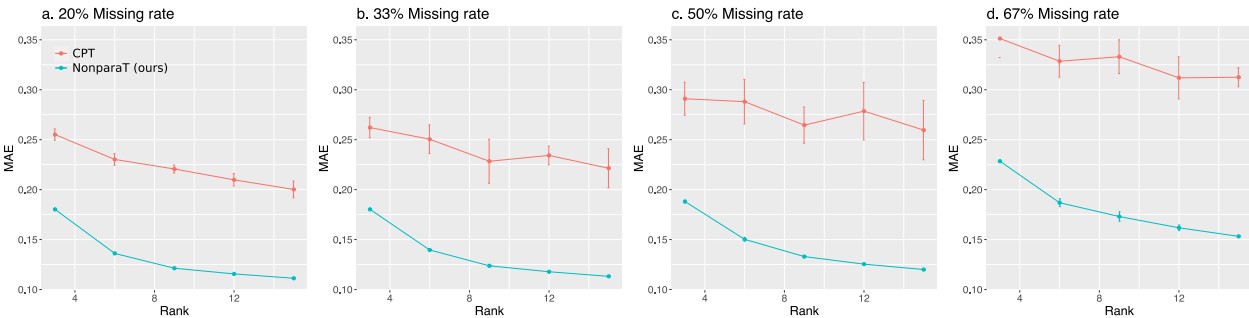

Supplementary Figure S3: Estimation error versus rank under different missing rate. Panels (a)-(d) correspond to missing rate 20%, 33%, 50%, and 67%, respectively. Error bar represents the standard error over 5-fold cross-validations.

We next investigate the pattern in the estimated signal tensor. Figure 4 of the main paper shows the identified top edges associated with IQ scores. Specifically, we first obtain a denoised tensor $\hat{\Theta} \in \mathbb{R}^{68 \times 68 \times 114}$ using our method with $r = 10$ and $H = 20$. Then, we perform a regression analysis of $\hat{\Theta}(i, j, :) \in \mathbb{R}^{144}$ against the normalized IQ score across the 144 individuals. The

regression model is repeated for each edge $(i, j) \in [68] \times [68]$. We find that top edges represent the interhemispheric connections in the frontal lobes. The result is consistent with the role of interhemispheric connectivity in human intelligence. The running times for performing one run on MRN-144 data is 5.1min evaluated on a single processor on an iMac (Mac OS High Sierra 10.13.6) desktop with Intel Core i5 (64 bit) 3.8 GHz CPU and 8 GB RAM.

## C.3 NIPS data analysis

In the main paper we have summarized the MAE in cross-validation experiments for $r = 6, 9, 12$. Here we provide additional results for a wider range $r = 3, 6, \ldots, 15$. Table S1 suggests that further increment of rank appears to have little effect on the performance. In addition, we also perform naive imputation where the missing values are predicted using the sample average. The two tensor methods outperform the naive imputation, implying the necessity of incorporating tensor structure in the analysis. The running times for performing one run on NIPS data is 4.4min evaluated on a single processor on an iMac (Mac OS High Sierra 10.13.6) desktop with Intel Core i5 (64 bit) 3.8 GHz CPU and 8 GB RAM.

| Method | $r = 3$ | $r = 6$ | $r = 9$ | $r = 12$ | $r = 15$ |
|---|---|---|---|---|---|
| NonparaT (Ours) | **0.18**(0.002) | **0.16**(0.002) | **0.15**(0.001) | **0.14**(0.001) | **0.13**(0.001) |
| Low-rank CPT | 0.22(0.004) | 0.20(0.007) | 0.19(0.007) | 0.17(0.007) | 0.17(0.007) |
| Naive imputation | | | 0.32(.001) | | |

Supplementary Table S1: Prediction accuracy measured in MAE in the NIPS data analysis. The reported MAEs are averaged over five runs of cross-validation, with standard errors in parentheses. Bold numbers indicate the minimal MAE among three methods. For low-rank CPT, we use R function `rTensor` with default hyperparameters, and for our method, we set $H = 20$.