# OpenReview forum: "Beyond the Signs: Nonparametric Tensor Completion via Sign Series"
_NeurIPS.cc/2021/Conference — NeurIPS 2021 Poster_

### Official Review · Reviewer_UqMu · 2021-07-17

**Rating:** 4
**Confidence:** 3

**Summary:**

This work studies the problem of tensor estimation from noisy observations with possibly missing entries. A nonparametric approach to sign tensor completion is developed. which can deal with both low-rank and high-rank tensor decomposition. Theoretical analysis is provided in terms of risk bound, estimation error, and sample complexity.


**Limitations And Societal Impact:**

Yes

**Main Review:**

The sign tensor decomposition problem is important, and the paper is clearly written However, the major concern lies in the lack in numerical tests.

The experiments were only done to compare with the basic low-rank tensor CP decomposition (CPT), and the matrix version of the proposed method was applied to tensor unfolding (NonparaM). The reviewer finds this inadequate and cannot support the effectiveness of the proposed algorithm. As CPT is a basic tensor decomposition algorithm that is for low-rand tensor decomposition, and the tensor-version outperforms the matrix version of the proposed model is also obvious. In order to improve the significance and novelty of the manuscript, more comparisons with state-of-art sign-tensor decomposition methods should be provided.

More discussion about the disadvantage and advantage of the theoretical analysis with respect to existing works on sign tensor decomposition is also required.

**Time Spent Reviewing:**

3

---

> ### Author Response · Authors · 2021-08-10
> **Response to Review 4**
>
>
> > The sign tensor decomposition problem is important, and the paper is clearly written.
>
> We appreciate the kind comment.
>
> >  More discussion about the disadvantage and advantage of the theoretical analysis with respect to existing works on sign tensor decomposition is required.
>
> Actually, we did present extensive theoretical comparison with existing works in Section 4.1 (page 7, entire section after L271). The original comparison was perhaps muddled in the technical description, so we summarize the key results here for easy reference.
>
>
> Model  | Our rate (power of $d$) | Comparison to state-of-art
>   :---|:---:|---:
>   Smooth tensor | $-(K-1)\min({\alpha\over \alpha+2}, {1\over 2})$| -
>   Single index model for $K=2$ | $-1/3$ | Improves previous rate $-1/4$ [15]
>   Single index model for $K\geq 3$ |  $-(K-1)/3$ |-
>   Tensor block model | $-(K-1)/2$ | Achieves minimax rate [37]
>   Generalized linear model |  $-(K-1)/3$| Close to minimax rate [39]
>
>
>
> As seen from the table, our theoretical analysis not only improves previous results for most well-studied tensor models, but also establishes the rates that were previously impossible. For instance, our theory shows error rate $d^{-(K-1)/3}$ when applying to single index tensor model. In the matrix case with $K=2$, our rate $d^{-1/3}$ is faster than previous rate $d^{-1/4} $ [Ganti et al, NeurIPS15]. The rate for single index model with arbitrary $K\geq 3$ was previously unknown, and we are among the first to establish convergence rate for this general model. In addition, our estimator is adaptive in the sense that if the data admits the parametric model (e.g tensor block model), then our method achieves an equally optimal minimax rate as parametric methods [Wang and Zeng, NeurIPS19]. Our result also improves earlier work [Ghadermarzy et al 2019, Lee and Wang 2020] by allowing both low- and high-rank signals. We show that the rate depends on only sign rank, which is provably often smaller than the usual tensor rank (Proposition 1, L138-139).
>
> We will add the above table to the paper for better exposition.
>
> > The major concern lies in the lack in numerical tests. The experiments were only done to compare with the basic low-rank tensor CP decomposition (CPT), and the matrix version of the proposed method was applied to tensor unfolding (NonparaM). The reviewer finds this inadequate and cannot support the effectiveness of the proposed algorithm. In order to improve the significance and novelty of the manuscript, more comparisons with state-of-art sign-tensor decomposition methods should be provided.
>
>
> Just to clarify, we did not attempt to propose a new tensor algorithm; instead, we present a learning reduction by using existing algorithms for a more challenging task (high-rank signal estimation). The nature of learning reduction procedure limits the choice of alternative algorithms to compare against. In particular, we exclude sign-tensor decomposition algorithms in the comparison because they have already been included as base algorithms in our learning-reduction pipeline. Comparing with them will lead to misleading (exaggerated) benefits of our method.
> In addition, many existing algorithms are special cases of our methods (e.g., single index tensor method, generalized linear tensor method). Our provable improvement in the above table provides a more convincing comparison than numerical tests.
>
>
> Inspired by reviewer's comment, we have implemented extra numerical comparison with three more algorithms: Tucker decomposition, CP decomposition under new R settings (denoted CPD2 in column 4) and averaged CP decomposition (denoted CPD3 in column 5). Here we provide numerical summary statistics for the newly added curve in figure 3(a)-(b); the full results will be incorporated in Figure 3(a)-(d) in final paper. We find that our method consistently outperforms both newly added algorithms. In addition, the Tucker and CP methods show similar performance. The possible reason is that CP and Tucker address low-rank structure, so their statistical accuracy suffers from practical nonlinear structures.
>
>
>
>
>  *Model 2 at d = 20,40,60:  All numbers are displayed on the scales $\times 10^{-2}$. Numbers smaller than $10^{-4}$ are shown as zero for space considerations.*
>
>   Tensor dimension  | NonparaT (ours) | CPD (default) |  CPD2 (new) | CPD3 (new)  |Tucker (new)
>   ------------------|-----------------|---------------|-----------------|----------|------
>   $d = 20$ | **5.05 (.16)** | 6.05 (.04) | 6.05 (.03) | 6.04 (0) |  6.06 (0)
>   $d = 40$ | **4.74 (.13)** | 5.66 (.01) | 5.66 (.01) | 5.65 (0)|  5.67 (0)
>   $d = 60$ | **4.68 (.11)** | 5.56 (.01) | 5.56 (0) | 5.56 (0) | 5.56 (0)
>
>
>
>   *Model 3 at d = 20,40,60:*
>
>   Tensor dimension | NonparaT (ours) | CPD (default) | CPD2 (new)|  CPD3 (new) | Tucker (new)
>   ------------------|-----------------|---------------|-----------------|----------|----------
>   $d = 20$ | **1.90 (.08)** | 2.68 (.04) |2.68 (.02) | 2.68 (0) | 2.67 (0)
>   $d = 40$ |  **1.33 (.05)** | 2.53 (.03) |  2.53 (.01) | 2.53 (0) | 2.52 (0)
>   $d = 60$ | **1.13 (.03)** | 2.51 (.01) | 2.49 (.01) |  2.49 (0) | 2.48 (0)
>
>
>
> Finally, we have also taken great care to make an accurate and fair comparison between the relevant methods discussed in the paper. In particular, Figure 3(a)-(c) compares parametric vs. non-parametric methods; Figure 3(b)-(d) compares matrix vs. tensor methods; Supplementary Figure S1(a)-(d) and S2(a)-(d) shows the comparison under four generating models; Supplementary Figure S3 compares the completion performance for Brain data; Supplementary Table S1 compares the estimation error for NIPS data. Our implementations may not be the most extensive possible, but they fairly compare competing algorithms on a wide range of aspects.

---

### Official Review · Reviewer_UX9r · 2021-07-18

**Rating:** 7
**Confidence:** 3

**Summary:**

This paper presents a nonparametric tensor completion technique that recovers a tensor that is not necessarily low-rank. At each target level to compare the entries to, it finds a low rank tensor that can approximate the sign tensor of difference between the original tensor and the level, and gets an approximation to the original tensor by averaging over multiple levels.

**Limitations And Societal Impact:**

Yes.

**Main Review:**

Strengths:

- This paper offers a new and provable perspective for low-rank and high-rank tensor completion, which should be very interesting to the tensor decomposition community.

- The main theorems are presented clearly in general.

- The results seem original.

Some questions:

- When partitioning [-1, 1] as $\mathcal{N}^c \cup \mathcal{N}$, how would $\rho(\pi, \mathcal{N})$ be defined if $\mathcal{N} = \emptyset$? That corresponds to the case in which $G(\pi)$ is flat across [-1, 1], and should be a case in which recovery is the easiest. So, if we define $\rho(\pi, \mathcal{N}) := \Delta s$ in this case, it is strictly smaller than $\mathcal{N} \neq \emptyset$, and the bound in Line 248 Equation 8 seems to be looser. This aspect does not seem to have been (intuitively) addressed either in the proof in Appendix A.3. Similarly, it might be helpful to show how theorems hold for corner cases.

- What would happen if we set the rank upper bound (which is a hyperparameter in the algorithm) in Line 240 Equation 7 to be either larger or smaller than the true $r$? Is it hard to show such results that the rank upper bound no smaller than $r$ would give a much better performance than smaller?

Some minor questions or suggestions:

- The loss L in Line 183 Equation 4 is simultaneously parameterized by $Z$, $Y$ and $\pi$. It might be clearer if the parameter $\pi$ can be added to the equation, to make it more evident that this is with respect to a single target level.

- I would suggest the authors show an illustrative figure of the definition of $G(\pi)$, $\mathcal{N}$ and $\mathcal{N}^c$, as I find it helps to understand the relevant notations that are central to this paper.

- Notation: In Appendix A.2 Equation 5, does $|\hat{\Theta}|$ refer to the entrywise norm (based on equations around Line 204 and 207 in the main paper)? It might be better to introduce this notation as well in the section in Line 77.

- This is certainly beyond the scope of this paper, but can we make the difference among target levels in $\mathcal{H}$ nonuniform? Namely, can we sample target levels more densely in $\mathcal{N}$ than in $\mathcal{N}^c$, so that we may get a better recovery with the same number of positive levels $H$?


**Time Spent Reviewing:**

4

---

> ### Author Response · Authors · 2021-08-10
> **Response to Review 3**
>
> >This paper offers a new and provable perspective for low-rank and high-rank tensor completion, which should be very interesting to the tensor decomposition community.
> The main theorems are presented clearly in general.
> The results seem original.
>
> Thanks for your positive support and helpful remarks! Below are our answers to your questions.
>
> > When partitioning $[-1,1]$ as $\mathcal{N}^c\cup \mathcal{N}$, how would $\rho(\pi,\mathcal{N})$ be defined if $\mathcal{N}=\emptyset$? That corresponds to the case in which $G(\pi)$ is flat across $[-1,1]$, and should be a case in which recovery is easiest. So, if we define $\rho(\pi,\mathcal{N}) := \Delta s$ in this case, it is strictly smaller than $\mathcal{N}\neq \emptyset$, and the bound in Line 248 Equation 8 seems to be looser. This aspect does not seem to have been (intuitively) addressed either in the proof in Appendix A.3. It might be helpful to show how theorems hold for corner cases.
>
> Your intuition is right, but we did not define $\rho(\pi,\mathcal{N})=\Delta s$ when $\mathcal{N} = \emptyset$. Instead, we use the convention to define $\rho(\pi,\mathcal{N}) = \infty$ when $\mathcal{N} = \emptyset$. We realize that the current notation may have caused the confusion; we will make clarification in the paper.
>
> With the above convention, an opposite direction is to be expected as in your premises -- Line 248 Equation 8 is sharper when $\mathcal{N}=\emptyset$.  In fact, we have presented exactly the same corner cases as you suggested in L276-284 (Example 3-4, GLMs/SIMs). We summarize the results here for easy reference; see Table below.
>
>
>
>  Model  | $\alpha$| $\vert\mathcal{N}\vert$| Our rate (power of $d$) | Previous result
>    :---|:---:|:---:|:---:|---:
>    Tensor block model | $\infty$| Finite| $-(K-1)/2$ | Achieves minimax rate [37]
>    Single index model for $K=2$ |1| Empty | $-1/3$ | Improves previous rate $-1/4$ [15]
>    Single index model for $K\geq 3$|1| Empty|   $-(K-1)/3$ |-
>    Generalized linear model |1 |  Empty |  $-(K-1)/3$| Close to minimax rate [39]
>    Structure with repeating entries |$\infty$| $\vert\mathcal{N}\vert$ |$-(K-2)/2$| -
>    Smooth tensor | $\alpha$ | - |  $-(K-1)\min({\alpha\over \alpha+2}, {1\over 2})$| -
>
>
> These examples instantiate our general theory to special models when $\mathcal{N}$ is either $\emptyset$, finite, or infinite. We find that, when $G(\pi)$ is flat (included in rows 2-4 in Table), our nonparametric rate achieves nearly minimax optimality as parametric rate. We will add the table to the final paper, space permitting.
>
>
> > What would happen if we set the rank upper bound in line 240 Equation 7 to be either larger or smaller than the true $r$. Is it hard to show such results that the rank upper bound no smaller than $r$ would give a much better performance than smaller?
>
> We interpret this question as the theoretical investigation of bias-variance trade-off determined by $r$. This is an important and challenging topic for nonparametric learning; we will discuss this question in our specific contexts.
>
>
> In our theory, we assume the true $r$ is known or otherwise has been consistently estimated; the adaptivity to unknown $r$ is empirically addressed by cross-validation. In principle, a larger $r$ leads to a smaller approximation bias but a larger estimation variance; vice versa for a smaller $r$. When we set the rank smaller than true signal rank $r$, this would incur extra approximation bias. The level of approximation bias depends on the spectral complexity of the signal tensor. In the matrix case, the approximation bias is well quantified by tail eigenvalues; however, such extension to higher-order tensors is known to be very challenging (partly due to lack of spectral theory).
>
> If one is willing to sacrifice generality by imposing extra assumptions (e.g. orthogonal decomposability of signal tensors), then we are able to establish the conjecture by the reviewer; that is, *the rank upper bound no smaller than $r$ would give a better performance than smaller* under slow spectral decay conditions. However, this investigation is at a cost of loss of generality (indeed, orthogonally decomposable tensor models are rather special in that their spectral properties are similar to matrices). We position our work as a bridge connecting (parametric) low-rank tensors and (nonparametric) high-rank tensors, so we choose to present the most general cases. We will leave the optimality of unknown rank as future research.
>
>
> > This is certainly beyond the scope of this paper, but can we make the difference among target levels in $\mathcal{N}$ nonuniform? Namely, can we sample target levels more densely in $\mathcal{N}$ than in $\mathcal{N}^c$, so that we may get a better recovery with the same number of positive levels $H$.
>
>
> Thank you for the great point! We have also thought about this when preparing for the work. Our theory continues to hold true for non-uniform levels, as long as level grids converge at a same rate. In principle, using non-uniform levels grid may improve the adaptivity, and we conjecture that the optimal level design depends on the true (but unknown) shape of density function $G(\cdot)$. Our simulation and data analysis, however, suggest little improvement of non-uniform designs. The explanation is that, in real applications, we usually have no information on function shapes such as $\mathcal{N}$, $\mathcal{N}^c$, $\alpha$, etc. The lack of prior knowledge makes non-uniform level designs sensitive to model misspecification. Therefore, we choose to present uniform design to achieve a good balance between theory and empirical applicability.
>
>
>
> > Some minor questions or suggestions: Notation in Line 183 Equation 4. Notation in Appendix A.2 Equation 5. I would suggest the authors show an illustrative figure of the definition of $G(\pi)$, $\mathcal{N}$, and $\mathcal{N}^c$, as I find it helps to understand the relevant notations that are central to this paper.
>
>
> Thank you for your constructive remarks that have greatly improve the paper! We have prepared the illustrative figure as you suggested, but seems OpenReview discourages the figure option. We will incorporate the edits in the final version.

---

### Official Review · Reviewer_cuu7 · 2021-07-19

**Rating:** 5
**Confidence:** 4

**Summary:**

The paper considers estimating a signal tensor from noisy and incomplete measurements. A new models is proposed based on sign-ranks of tensors. Statistical estimation rates are provided (which pretty much follows standard template). Algorithms and extensive numerical results are provided selling the proposed model.



**Ethical Concerns:**

none (to my knowledge)

**Ethics Review Area:**

["I don’t know"]

**Limitations And Societal Impact:**

none (to my knowledge)

**Main Review:**

1) I am not sure if the results in figure 1(a) provide sufficient motivation. It is folklore results in tensor community that CP ranks are unstable in various sense and Tucker ranks provide a more stable notion of rank for higher-order tensors. Is there any reason why CP rank is used for motivation (not in the form of citation to prior works) ? Can the authors provide a plot of c versus Tucker rank in this experimental setup ?  More importantly, even for the provided result, there is a big discrepancy between the actual CP rank and the numerical computation due to the computational complexity results.

2) The second example provided in lines 41-45 is closely related to hypergraphons (https://arxiv.org/pdf/1302.1634.pdf). This should be discussed carefully.

3) The statement in line 63-64 is not clear and perhaps hiding a lot of things. For example, in the example considered in [20, Theorem 4.1], they require the condition on the signal-to-noise ratio (i.e., the term \bar{\lambda}/\sigma) to obtain the initializer. Only in this high signal-to-noise ratio regime, polynomial time algorithms exists. Similarly, in [36, page 15], it is not at all clear what the order T_0 is. It is well know that most tensor problems have a statistics-computation gap and proving no algorithms exists in this gap regime is (one of the) most interesting problem with tensor estimation/learning.

4) Regarding Theorem 4, how is (27) solved ? From the way the results are presented, it is not clear what the signal-to-noise ratio of the problem is.

5) Given all the above issues (unclear parts, at least to this reviewer) the paper looks like it is proposing yet another model for tensor problems. Agreed, some models might be useful to some (practical) problems and I do see some merits in the proposed model for some applications. Hence, I am on the fence with respect to this paper.

**Time Spent Reviewing:**

10 hours

---

> ### Author Response · Authors · 2021-08-11
> **Response to Review 2**
>
> We thank the reviewer for the constructive suggestions. Below are our detailed response.
>
> > I am not sure if the results in figure 1(a) provide sufficient motivation. It is folklore results in tensor community that CP ranks are unstable in various sense and Tucker ranks provide a more stable notion of rank for higher-order tensors. Is there any reason why CP rank is used for motivation (not in the form of citation to prior works)? Can the authors provide a plot of c versus Tucker rank in this experimental setup? More importantly, even for the provided result, there is a big discrepancy between the actual CP rank and the numerical computation due to the computational complexity results.
>
> There seem to be two questions here, one relating to motivation of Figure 1, and the other relating to CP rank. We address the two questions separately.
>
> Our Figure 1 shows that nonlinear transformation does not preserve low-rankness of original tensor. Same observation applies to *both* CP and Tucker models, and more generally, to all low-rank models with scale-sensitive rank measures. We have repeated the experiment in Figure 1, and found the same pattern regardless of CP or Tucker model (see Table below; more on this point in next paragraph).
>
>
> The observed rank non-stability motivates us to develop nonparametric sign measure, for which the key benefit comes from the reduction:
> \begin{equation}
> \text{intrinsic nonlinear complexity}\lesssim \text{sign measure}
> \stackrel{(i)}{\ll}  \text{CP rank (measured by $r$)}  \stackrel{(ii)}{\leq}  \text{Tucker rank (product of $r_k$)}.\tag{1}
> \end{equation}
>
> It is helpful to realize that CP rank is always a more conservative complexity measure than Tucker rank (see (ii)). In other words, our Figure 1 on motivating inequality (i) directly implies the benefits over Tucker model. Because (i) is a sharper inequality, we choose to plot CP rank as a motivation.
>
> Regarding the comments on CP rank: We agree with the reviewer that actual CP rank is impossible/unstable to compute. This is precisely what we have pointed out in Appendix B.1. However, the discrepancy between the actual and numerical CP ranks does not undermine our premises; in fact, it provides a more convincing reason for motivating our work. The 90%-approximated CP rank reported in Figure 1 is a conservative *lower bound* of actual unknown CP rank. The observed rank non-stability highlights the weakness of usual CP model.
>
>
> In our case, both CP and Tucker models show equally poor performance under nonlinear transformation. We have run extra simulation (see Table below) to compare the estimation error with Tucker model in L325-326. Just to clarify, for the reviewer's interest, both Tucker and CP models have their own plus and minus. While it is straightforward to calculate $\varepsilon$-approximated Tucker rank $r_k$ from unfolding a tensor along each mode, the assembled $(r_{1},\ldots,r_{K})$-Tucker tensor does not admit $\epsilon$-approximation anymore. In fact, earlier work has shown the NP-hardness of Tucker estimation under certain signal-to-noise regime [Zhang and Xia 2018, Wang and Li 2020, Han et al 2020]. Our implementation is under default setting of built-in `tucker` function in `rTensor` package.
>
>
>   *Model 2 at d = 20,40,60:  All numbers are displayed on the scales $\times 10^{−2}$. Numbers smaller than $10^{-4}$ are shown as zero for space considerations.*
>
>   Tensor dimension  | NonparaT (ours) | CPD (default) |   Tucker (new)
>   ------------------|-----------------|---------------|-----------------
>   $d = 20$ | **5.05 (.16)** | 6.05 (.04)  |  6.06 (0)
>   $d = 40$ | **4.74 (.13)** | 5.66 (.01) |  5.67 (0)
>   $d = 60$ | **4.68 (.11)** | 5.56 (.01) | 5.56 (0)
>
>
>
>   *Model 3 at d = 20,40,60:*
>
>   Tensor dimension | NonparaT (ours) | CPD (default) |  Tucker (new)
>   ------------------|-----------------|---------------|-----------------
>   $d = 20$ | **1.90 (.08)** | 2.68 (.04) |2.67 (0)
>   $d = 40$ |  **1.33 (.05)** | 2.53 (.03) |   2.52 (0)
>   $d = 60$ | **1.13 (.03)** | 2.51 (.01) | 2.48 (0)
>
> >The second example provided in lines 41-45 is closely related to hypergraphons. This should be discussed carefully.
>
> We thank the reviewer for the reference. We are aware of the work, and we emphasize that our example should *not* be confused with hypergraphon. We will add the following discussion to the paper for completeness:
>
> "Our example should not be confused with hypergraphons [Zhao 2014, Lovasz and Szegedy 2006]. Hypergraphon is a limiting function based on a sequence of uniform hypergraphs in cut distance. Unlike the matrix case where graphon is represented as a bivariate function, hypergraphons for order-$K$ tensors should be represented as a $(2^{K}-2)$-variate function [Zhao 2014, Section 1.2]. Our example differs from hypergraphon from at least three aspects: i) our function representation uses $K$ variables not $(2^{K}-2)$; ii) we impose no randomness on design points $x_{i}$; iii) our function is used only for concise notation for tensor entry values. Whether it is possible to extend our theory to hypergraphon is an interesting question for future research."
>
>
> >The statement in line 63-64 (Introduction) is not clear and perhaps hiding a lot of things. For example, in the example considered in [20, Theorem 4.1], they require the condition on the signal-to-noise ratio (i.e., the term $\bar{\lambda}/\sigma)$ to obtain the initializer. Only in this high signal-to-noise ratio regime, polynomial time algorithms exists. Similarly, in [36, page 15], it is not at all clear what the order $T_0$ is. It is well know that most tensor problems have a statistics-computation gap and proving no algorithms exists in this gap regime is (one of the) most interesting problem with tensor estimation/learning.
>
>
> We agree with the reviewer that the success of (or the lack of) polynomial algorithm relies on the signal-to-noise ratio. We do emphasize in our Section 4.2 that *technical assumptions on base algorithm* (L311) needs to be satisfied in order to establish computational complexity. We will also add this statement in Introduction, L63-64 "a number of polynomial-time algorithms are readily available *under moderate-to-high signal-to-noise ratio conditions*."
>
>
> A major point we want to make in response is that, we did not attempt to propose a new tensor algorithm; instead, we present a *learning reduction* by adopting existing algorithms for a more challenging task (high-rank signal estimation). Our algorithmic complexity is $\text{poly}(d)$ times that of base algorithm. Our problem shares the same phase-transition (in terms of statistical-computational efficiency) as the base algorithm. The notion of signal-to-noise ratio depends on the specific choice of base algorithms. As the reviewer mentioned, when 1-bit tensor estimation algorithms in [36,20] are used as base algorithm, we need the signal-to-noise requirement (spectral complexity of signal tensor $\Theta$ and corresponding noise) for polynomial complexity. This assumption is what we imposed in Section 4.2.
>
> To eliminate the possible confusion, we will add the following remarks to Theorem 4: *technical assumptions depend on the chosen algorithm. A signal-to-noise ratio for base problem is needed for the polynomial complexity of algorithms in [36,20].*
>
>
>
> >Regarding Theorem 4, how is (27) solved? From the way the results are presented, it is not clear what the signal-to-noise ratio of the problem is.
>
> The formulation (27) shares similar formulation as (4) in [5] and Section 4.4 in Han et al 2021 (arxiv: 2002.11255). We implement (27) in our submitted code using alternating optimization with spectral initialization. Other choices of base algorithms are also possible.
> The exact formula of signal-to-noise ratio depends on the chosen algorithm. For example, convex relaxation [17] defines the signal-to-noise ratio by $|\Theta|_{\max}/\sigma$; projected gradient descent [Han et al 2021] defines the signal-to-noise ratio by $\bar \lambda/\sigma$. We do not confine our framework to a certain developed algorithm but allow users to choose their own favorite algorithm adapting the algorithmic guarantees in our context.
>
>
>
>  >Statistical estimation rates are provided (which pretty much follows standard template)
>
>  We would like to emphasize our technical contribution is beyond standard template. The following table summarizes our statistical rate compared to existing work (see Section 4.1, entire section after L271).
>
>
>   Model  | Our rate (power of $d$) | Previous results
>    :---|:---:|---:
>    Smooth tensor | $-(K-1)\min({\alpha\over \alpha+2}, {1\over 2})$| -
>    Single index model for $K=2$ | $-1/3$ | Improves previous rate $-1/4$ [15]
>    Single index model for $K\geq 3$ |  $-(K-1)/3$ |-
>    Tensor block model | $-(K-1)/2$ | Achieves minimax rate [37]
>    Generalized linear model|  $-(K-1)/3$| Close to minimax rate [39]
>
>
>
>  As seen from the table, our theoretical analysis not only improves previous results for many well-studied tensor models, but also establishes the rates that were previously impossible. For instance, our theory shows error rate $d^{-(K-1)/3}$ when applying to single index tensor model. In the matrix case with $K=2$, our rate $d^{-1/3}$ is faster than previous rate $d^{-1/4} $ [Ganti et al, NeurIPS15]. The rate for single index model with arbitrary $K\geq 3$ was previously unknown, and we establish the statistical rate for this general model. In addition, we find that our nonparametric rate achieves nearly minimax optimality as parametric rate when signal tensor admits low-rankness (e.g., tensor block model).
>
> We leverage nonparametric tools to derive simple but sharper rate than previous possible methods (see the summarized table above). Considering the incisive minds that have studied tensor completion and nonparametric function estimation separately, we view it as a strength of the paper providing such connections that were not previously formulated.

---

> > ### Comment · Reviewer_cuu7 · 2021-08-25
> > **Reply to response**
> >
> > Thanks for the response.
> >
> > Hypergraphons: Please note that your model is indeed a special case of general hypergraphons. Indeed, as it depends only on $K$ coordinates, it is referred to as simple hypergraphons (see e.g., https://link.springer.com/article/10.1023/A:1021692202530). And you require uniform (and non-uniform) sampling for observed entries.
> >
> > Rates: Generalized linear models are invariably a special case of single index models, aren't they? Also, for the SIM with more general link functions, much improved rates are also obtained, for example, in http://proceedings.mlr.press/v80/xu18a/xu18a.pdf. See also http://proceedings.mlr.press/v80/xu18a/xu18a.pdf for related work in the matrix setting, both of which are not discussed.
> >
> > Computation-Statistics Tradeoff: As mentioned in the review and by the authors in the response, their (algorithmic) proposal depends on the signal-noise ratio assumptions as in previous works. This is my main concern. To summarize: A model is proposed and statistical estimation rates are established. However, the implementable estimator would work provably only in the high signal-to-ratio regime. This story has been discovered and quantified in the tensor context (other estimation contexts) over the last decade. Hence, I am inclined to view the proposed model as yet another model fitting this developed story. (However, as I told, of course it might have some applications and I am not an expert to judge the practical applicability of the model).

---

> > > ### Author Response · Authors · 2021-08-26
> > > **Thank you for helpful discussion**
> > >
> > > > Hypergraphons: Please note that your model is indeed a special case of general hypergraphons. Indeed, as it depends only on $K$ coordinates, it is referred to as simple hypergraphons (see e.g., https://link.springer.com/article/10.1023/A:1021692202530). And you require uniform (and non-uniform) sampling for observed entries.
> > >
> > > We thank the reviewer for mentoring the work of hypergraphons [Kallenberg 1999] and matrix graphons [Xu 2017]. Both topics are certainly relevant, and should be discussed in details.
> > >
> > > Hypergraphon: Our original intent is to avoid overselling our work, so we choose to make careful distinctions about our model vs. general hypergraphons. Regarding *"...(our) model is a special case of simple hypergraphons."* This is actually not what we intended to convey. In fact, we have shown the opposite way---*"simple hypergraphon is a special case of our model."* We use simple hypergaphon as one of five concrete examples to illustrate our general sign-representable tensors. The illustration, however, is not intended in any way to restrict the generality of our model (L146-153).
> > >
> > > We agree with reviewer that discussion on hypergraphon is helpful. We will add the following paragraph to Related Work:
> > >
> > > "Our last example is related to hypergraphons [Zhao 2014, Lovasz and Szegedy 2006]. Hypergraphon is a limiting function based on a sequence of uniform hypergraphs in cut distance. Though hypergraphon is an important application, the implication of our results should be interpreted with cautions for two reasons: 1) unlike the matrix case where graphon is represented as a bivariate function, general hypergraphons for order-$K$ tensors should be represented as $(2^K-2)$-variate function [Zhao 2014, Section 1.2]. Our example depends on $K$ coordinates only, and in this sense, it shares the common ground as simple hypergraphons [Kallenberg 1999]. 2) Unlike typical simple hypergraphons where the design points are random variables $x_i\sim \text{Uniform}[0,1]$, our example uses deterministic design points $x_i=i/d$. These two choices lead to a notable difference in the RMSE rate $d^{-(K-1)/3}$ (ours) vs. $d^{-1}$ (simple hypergraphon) [Krishnakumar 2021 (appeared after our NeurIPS submission)]. We see that our model has a substantially faster rate than simple hypergraphon as tensor order $K\to \infty$. This improvement stems from the distinction of fixed vs. random designs. In simple hypergraphon models, the error is dominated by the randomness in design points measured by $\text{dist}(x_i,x_j)$, which has a standard error at least $d^{-1}$ [2-4]. In contrast, we use grid points only for the purpose of representing tensor entries, and no randomness is involved in designs. Whether it is possible to extend our theory to general hypergraphon is an interesting question for future research."
> > >
> > > > Rates: Generalized linear models are invariably a special case of single index models, aren't they? Also, for the SIM with more general link functions, much improved rates are also obtained, for example, in http://proceedings.mlr.press/v80/xu18a/xu18a.pdf. See also for related work in the matrix setting, both of which are not discussed.
> > >
> > > We thank the reviewer for mentioning related work on matrices. The graphon paper by Xu (2017) is important, and is in line with several matrix papers in our original references [e.g. 8, 13-15]. These matrices papers indeed have motivated us to develop the current work on tensors. Xu (2017) established the RMSE rate $d^{-1/3}$ for Lipschitz bivariate functions. This is precisely the same rate in our general theorem by setting tensor order $K=2$, thereby suggesting the sharpness of our extension. We thank the reviewer for drawing our attention the suboptimality of Ganti (2015) compared to Xu (2017), and we will update this reference throughout the paper.
> > >
> > > We will add following discussion to Related Work.
> > >
> > > "High-rank matrix estimation has been extensively studied under graphon models [Xu 2017, Zhang 2017, Chan 2014], nonlinear models [15], and subspace clustering [32, 13]. When we set tensor order $K=2$, our sign-representable framework extends the function family in Xu (2017). In particular, we consider $\alpha$-smooth cumulative distribution function (CDF) $G\colon \mathbb{R}\to \mathbb{R}$ in the range space; this is in contrast to earlier (hyper)graphon models that consider $\alpha$-smooth functions $f\colon \mathbb{R}^{d\times \cdots\times d}\to\mathbb{R}$ in the domain space. Our benefit bears the analogy of Lebesgue versus Riemann integrals in functional analysis, in the sense that our neighborhood is determined by the range space not the domain space. The former approach is especially appealing for higher-order tensors because the range space is simple and 1-dimensional, whereas the domain space is huge and high dimensional.
> > >
> > > ...Our earlier example has shown the nonparametric rate $d^{-(K-1)/3}$ when applying our method to single index tensor model. In the matrix case with $K = 2$, our result yields the error rate $d^{-1/3}$, consistent with the RMSE rate obtained by Xu (2017). It has been widely conjectured that $d^{-1/3}$ is the best possible rate for $K=2$ among computationally efficient estimators (Xu 2017, Zhang 2017). We conjecture the similar phenomenon extends to higher-order tensors."
> > >
> > > For reviewer's interest, we have also added numerical comparisons with matrix graphon method. The following table shows that our tensor method improves over graphon methods. The outperformance is to be expected: our nonparametric tensor method can be viewed as---in some sense---a higher-order extension of graphon, and the improvement stems from the incorporation of higher-order structure in estimation.
> > >
> > >   *Estimation error for Model 3 at d = 20,40,60:*
> > >
> > >   Tensor dimension | NonparaT (ours) | CPD (default) | Graphon (new)
> > >   ------------------|----------------- |----|--|
> > >   $d = 20$ | **1.90 (.08)** | 2.68 (.04)  | 2.11 (.02)
> > >   $d = 40$ |  **1.33 (.05)** | 2.53 (.03) | 1.62 (.02)
> > >   $d = 60$ | **1.13 (.03)** | 2.51 (.01)  |  1.43 (.01)
> > >
> > >
> > >   *Estimation error for Model 4 at d = 20,40,60:*
> > >
> > >   Tensor dimension | NonparaT (ours) | CPD (default) | Graphon (new)
> > >   ------------------|----------------- |----|--|
> > >   $d = 20$ | **3.75 (.43)** | 5.52 (.04)  | 5.64 (.10)
> > >   $d = 40$ |  **2.84 (.24)** | 4.63 (.01) | 3.66 (.02)
> > >   $d = 60$ | **2.30 (.09)** | 4.38 (.01)  |  3.10 (.02)
> > >
> > >
> > > > Computation-Statistics Tradeoff: As mentioned in the review and by the authors in the response, their (algorithmic) proposal depends on the signal-noise ratio assumptions as in previous works. This is my main concern. To summarize: A model is proposed and statistical estimation rates are established. However, the implementable estimator would work provably only in the high signal-to-ratio regime. This story has been discovered and quantified in the tensor context (other estimation contexts) over the last decades. Hence, I am inclined to view the proposed model as yet another model fitting this developed story. (However, as I told, of course it might have some applications and I am not an expert to judge the practical applicability of the model).
> > >
> > > We agree with the reviewer that our work falls in the general vein of tensor model estimation. The lack of polynomial-time estimator in low signal-to-noise regime is an *inevitable fact*. We view this "negative" result as a sanity check that every reasonable estimation procedure (including ours) should respect. Indeed, the computational-statistical gap is a *provable* property in low-rank tensor problems [Cai 2021, Zhang 2018]. There is *no hope for a win-win solution to this problem*, because our general (high-rank) tensor model should incorporate low-rank models as special cases.
> > >
> > > We would position our work as developing a general tensor model that bridges parametric, low-rank tensors and nonparametric, high-rank tensors. As the reviewer said, there is a large literature on (downstream) analysis of tensor estimation properties. We try to avoid reinventing the wheel, so instead we offer a (upperstream) modeling solution. We develop a probable sign-representable tensor model that unifies low-rank and high-rank tensors. Our learning reduction strategy empowers existing algorithms for broader implication, thereby addressing more challenging high-rank problems almost for free (at only an extra poly$(d)$ computational cost, but no extra statistical cost). Considering the incisive minds that have studied tensor completion and functional estimation separately, we view it as a strength of the paper providing such connections that were not previously established. We hope the work opening up new inquiry that allows more researchers to contribute to this field.
> > >
> > >
> > > [1] Zhao, Yufei. Hypergraph limits: a regularity approach. Random Structures & Algorithms 47, no. 2: 205-226, 2015.
> > >
> > > [2] Balasubramanian, Krishnakumar. Nonparametric Modeling of Higher-Order Interactions via Hypergraphons. arXiv preprint arXiv:2105.08678, 2021.
> > >
> > > [3] Wei S, Madrid-Padilla OH, Sharpnack J. Distributed Cartesian Power Graph Segmentation for Graphon Estimation. arXiv preprint arXiv:1805.09978, 2018.
> > >
> > > [4] Zhang, Yuan, Elizaveta Levina, and Ji Zhu. Estimating network edge probabilities by neighbourhood smoothing. Biometrika. 104(4):771-83, 2017.
> > >
> > > [5] Cai, Changxiao, Gen Li, H. Vincent Poor, and Yuxin Chen. Nonconvex low-rank tensor completion from noisy data. Operations Research, 2021.
> > >
> > > [6] Zhang, Anru, and Dong Xia. Tensor SVD: Statistical and computational limits. IEEE Transactions on Information Theory 64, no. 11: 7311-7338, 2018.

---

### Official Review · Reviewer_uYrq · 2021-07-20

**Rating:** 6
**Confidence:** 4

**Summary:**

The authors proposed a decomposition method which represents a tensor as sum of a series of sign tensors, $\text{sign}(Z)$. Each tensor, $Z$, is estimated by minimizing a weighted classification objective function with the rank constraint, $rank(Z) <= r$, e.g., using 1-bit tensor estimation algorithms.


**Limitations And Societal Impact:**

The main concern is Comparison.
The proposed method may be promising but comparison is not good.


**Main Review:**


- The authors present two examples to demonstrate the motivation of the proposed model. According to the paper, the canonical polyadic tensor decomposition (CPD) fails in finding low-rank approximation of tensors obtained from low-rank tensors after nonlinear transforms.

First the nonlinear transforms do not preserve the low-rank structure of the original tensor, hence it is natural that the generated tensor can have higher rank. The data simply does not admit the CP model.

Second, even the transformed data can have higher rank than that of the original data, the numerical ranks at the relative approximation error of 0.1, shown in Figure 1, are not optimal, too high, perhaps due to incorrect decomposition. The authors do not explain the algorithm for CPD and its implementation used in their experiments.

For example the authors generated tensors of size $30 \times 30 \times 30$ $Z = {\bf a}^{\otimes3} + {\bf b}^{\otimes3} + {\bf c}^{\otimes3}$ where ${\bf a}$, ${\bf b}$, ${\bf c}$ consist of $N(0,1)$ entries.
The transformed tensor $\Theta = f(Z)$ can be approximated by a CP model with a rank of 60 with a relative approximation error smaller than $\frac{\|\Theta - \hat{\Theta}\|_F}{\|\Theta\|_F}<0.1$ for the case $c = 200$. This numerical rank is much smaller than the rank of 150 that the authors show in Figure 1.a.

For c = 50, the tensor $\Theta$ can have a numerical rank of 47, which is much smaller than the rank of 130 determined by the authors.

This is to show that the authors do not achieve the best results for CPD. The results for two examples in Section 5 may face the same problem, and the conclusions are not very accurate or bias.


- The proposed method approximates the tensor by a sum of $H = 20$ (or even more) low-rank tensors. Each estimated tensor can be low rank but not the mean tensor of low-rank tensors.

For a fair comparison, the authors should do the CPD completion method similarly, the final result should be mean of low-rank tensor approximated by CPD.

In summary, the obtained results using CPD are not optimal as shown above for Example1, and
and should be estimated as mean of multiple CP models.

Other comments

CPD (not CPT) is used for abbreviation of Canonical polyadic decomposition.

**Time Spent Reviewing:**

48

---

> ### Author Response · Authors · 2021-08-10
> **Response to Review 1**
>
> We thank the reviewer for the positive supports and helpful remarks. Below are our answers to your questions.
>
> > First the nonlinear transforms do not preserve the low-rank structure of the original tensor, hence it is natural that the generated tensor can have higher rank. The data simply does not admit the CP model.
>
> We respectfully point out this is the very argument from which our work is motivated. It is not surprising that classical CP method fails when data undergoes nonlinear transformations. What is surprising is that a simple but careful ensemble learning method (via sign tensor series as we proposed) makes the estimation *provably* robust to unknown transformations, thereby effectively addressing the aforementioned challenges. Classical CPD achieves accuracy only when the data admits CP model; whereas our model achieves accuracy under *both* CP models and a wide range of non-CP models.
>
>
>
> > Second, even the transformed data can have higher rank than that of the original data, the numerical ranks at the relative approximation error of 0.1, shown in Figure 1, are not optimal, too high, perhaps due to incorrect decomposition. The authors do not explain the algorithm for CPD and its implementation used in their experiments.
>
>
> We appreciate the working examples provided by the reviewer, and we are also able to reproduce the same calculation as the reviewer.
>
> With that said, our reported numbers in Figure 1(a) in Introduction section are also correct. We use default setting of built-in `cp` function (iteration = 25, tolerance = $10^{-5}$) in `rTensor` R package. Figure 1(a) serves as a motivating example where the low-rank model fails to represent tensors obtained from nonlinear transformation. The numbers we reported are summary statistics averaged over random rank-3 tensors from Monte Carlo simulations. We agree that, for some particular instances of random tensors and/or different precision configurations of R, the numerical rank increment may show up from 3 to $\approx$ 60, as calculated by the reviewer. In fact, this result *does* support our conclusion in Introduction:
>
> 1) Usual tensor rank is highly sensitive to nonlinear transformation, leading an rank increment from 3 (ground truth) to 60$\sim$150 (with numerical instability coming from randomness in signals, software configurations, optimization choices, etc);
>
> 2) The rank increment is also highly sensitive to particular instances of signal tensors, despite of a common intrinsic rank $=3$ to start with.
>
> Both observations highlight the weakness of usual CP rank, suggesting the need for a new complexity measure invariant to unknown nonlinear transformation across all tensor instances. This has motivated us to develop robust high-rank tensor models. For easier reproducibility, we will add more instructions in our submitted code. We will also add more implementation details to the current section Appendix B.1 "Sensitivity of tensor rank to monotonic transformations".
>
>
> > This is to show that the authors do not achieve the best results for CPD. The results for two examples in Section 5 may face the same problem, and the conclusions are not very accurate or bias.
>
>
> The CPD examples in Section 5 are calculated either from algebraic calculation (Table 1) or from cross validation (Figure 3). The rank in Table 1 is proved using algebraic equations (see Appendix Section B.2) for specially-structured tensors; therefore, no numerical approximation is involved. The prediction performance (Figure 3 and Table 2) calibrates the rank selection based on cross validation prediction error. We set the rank of CPD to be the one that yields the best prediction performance over $r=1,2,\ldots$. Therefore, the conclusions and comparison are valid.
>
>
> For reviewer's interest, we have added extra numerical experiments and verified that different implementations of CPD barely affect the results in Section 5. The following two tables summarize the estimation error under new R settings (denoted "CPD2", iteration = $10^3$, tolerance = $10^{-5}$, so as to mimic reviewer's working examples) based on Models 2-3 in Table 1 (L325-326). We compare the summary statistics in means and standard errors between CPD and CPD2. The results show that CPD2 yields similar performance as default CPD except reducing standard errors. Both implementations of CPD exhibit worse performance than our nonparametric models. This observation suggests that, compared to the algorithmic error, the statistical error is more likely to dominate in practice.
>
>  *Model 2 at d = 20,40,60:  All numbers are displayed on the scales $\times 10^{−2}$. Numbers smaller than $10^{-4}$ are shown as zero for space considerations.*
>
>   Tensor dimension  | NonparaT (ours) | CPD (default) |  CPD2 (new) | CPD3 (new)  |Tucker (new)
>   ------------------|-----------------|---------------|-----------------|----------|------
>   $d = 20$ | **5.05 (.16)** | 6.05 (.04) | 6.05 (.03) | 6.04 (0) |  6.06 (0)
>   $d = 40$ | **4.74 (.13)** | 5.66 (.01) | 5.66 (.01) | 5.65 (0)|  5.67 (0)
>   $d = 60$ | **4.68 (.11)** | 5.56 (.01) | 5.56 (0) | 5.56 (0) | 5.56 (0)
>
>
>
>   *Model 3 at d = 20,40,60:*
>
>   Tensor dimension | NonparaT (ours) | CPD (default) | CPD2 (new)|  CPD3 (new) | Tucker (new)
>   ------------------|-----------------|---------------|-----------------|----------|----------
>   $d = 20$ | **1.90 (.08)** | 2.68 (.04) |2.68 (.02) | 2.68 (0) | 2.67 (0)
>   $d = 40$ |  **1.33 (.05)** | 2.53 (.03) |  2.53 (.01) | 2.53 (0) | 2.52 (0)
>   $d = 60$ | **1.13 (.03)** | 2.51 (.01) | 2.49 (.01) |  2.49 (0) | 2.48 (0)
>
> > The proposed method approximates the tensor by a sum of $H=20$ (or even more) low-rank tensors. Each estimated tensor can be low rank but not the mean tensor of low-rank tensors. For a fair comparison, the authors should do the CPD completion method similarly, the final result should be mean of low-rank tensor approximated by CPD.
>
>
> We have run additional numerical experiments to implement the alternative CPD suggested by the reviewer (denoted CPD3 in column 5 of the above two tables). We took $H=20$ low-rank tensors from CPD and used the entrywise average as the estimate. The results confirm that entrywise average of low-rank tensors shows no improvements in performance but destroys the intrinsic low-rankness in the data. In contrast, our proposed sign series approach achieves accurate estimation under both linear and nonlinear models. In particular, the sign-series averaged tensor maintains low-rankness if the data admits the low-rankness, while gaining provable accuracy over existing methods when data violates the low-rankness. This comparison shows the outperformance of our adaptive nonparametric method over a wide range of existing low-rank models.

---

### Decision · Program_Chairs · 2021-09-27

**Decision:**

Accept (Poster)

**Comment:**

This paper presents an interesting and innovative new notion of tensor rank
that can model high rank tensors with low complexity,
and show how to use this idea to solve tensor completion problems.
While the reviewers differ in their beliefs about the impact this method will have,
none dispute that the paper is clear, interesting, correct, and even (mostly) complete,
after the new experiments the authors added in the discussion.
This paper is a novel and solid contribution that deserves to be published.